# Importance of basement faulting and salt decoupling for the structural evolution of the Fars Arc, Zagros fold-and-thrust belt: A numerical modeling approach

Fatemeh Gomar[1], Jonas B. Ruh[2], Mahdi Najafi[3], Farhad Sobouti[1]

[1]Department of Earth Sciences, Institute for Advanced Studies in Basic Sciences, Zanjan, 45137-66731, Iran
[2]Institute of Marine Sciences, Consejo Superior de Investigaciones Científicas, 08003, Barcelona, Spain
[3]Geosciences Barcelona, Consejo Superior de Investigaciones Científicas, GEO3BCN-CSIC, 08028, Barcelona, Spain

*Correspondence to: Fatemeh Gomar (fatemehgomar@iasbs.ac.ir) and Jonas B. Ruh (jruh@icm.csic.es)*

**Abstract.** Understanding the tectonic evolution and crustal-scale structure of fold-and-thrust belts is crucial for exploring geological resources and evaluating seismic hazards. We conducted a series of two-dimensional finite difference thermo-mechanical numerical models with a visco-elasto-plastic/brittle rheology to decipher how the interaction of inherited basement faults and salt décollement levels control the deformation process and structural style of the Fars Arc in the Zagros fold-and-thrust belt during tectonic inversion. Numerical experiments with extension and consequent convergence phases indicate that strain accumulation patterns during initial rifting is controlled by the location and geometry of prescribed faults. During convergence, the inverted basement faults form large-wavelength foreland-verging fault-propagation anticlines in the sedimentary cover, while the thick salt layer promotes the growth of second-order detachment anticlines accompanied by both fore- and back-limb thrust faults. Experiments without prescribed basement faults result in dispersed brittle/plastic deformation during rifting and convergence and an effective mechanical decoupling along the salt horizon. Overall, reactivated faults can serve as pathways for stress transfer, resulting in the formation of new faults triggering seismic activity. The structural evolution of orogenic belts like the Zagros do not adhere to a fixed pattern; it is shaped by factors such as basement rock properties and inherited fault orientations. Shallow earthquakes predominantly occur along décollement anticlines in the sedimentary cover, while deeper, larger ones are associated with basement faults. We also observe variations in resistance to deformation based on salt rheology and fault geometry, with listric faults minimizing resistance. The degree of basement involvement directly influences the model's resistance to deformation, with greater involvement facilitating easier deformation. Our results related to the temporal and spatial relationship between thin- and thick-skinned tectonics can work as analogues for similar orogenic belts worldwide, such as Taiwan, the Pyrenees, the Alps, the Appalachians, and the Kopet Dagh.

## 1 Introduction

Fold-and-thrust belts are complex tectonic domains formed in response to near- or far-field compressional stress fields in the Earth's crust. Multiple factors have been identified to control the structural styles in these belts, such as; the strength of the involved rocks (Davis et al., 1983), the involvement of the crystalline basement and its pre-existing faults (Pfiffner, 2017; Barchi and Tavarelli, 2022), the occurrence and rheology of décollement layers (Ruh et al.,

2012; Pla et al., 2019; Eslami Rezaei et al., 2023), and the intensity of surface processes (Talbot and Alavi, 1996; Cooper, 2007; Simpson, 2010; Malavieille and Konstantinov, 2010;  Morley et al., 2011; McQuarrie and Ehlers, 2017). In particular, the presence of décollements, mechanically weak layers, composed of rocks that have lower mechanical strength than their surroundings, plays a significant role in the formation and evolution of fold-and-thrust belts. These layers are weak stratigraphic horizons that separate and mechanically decouple layers of greater strength

and tend to accommodate and localize the deformation (Koyi and Mansurbeg, 2021). The relative weakness of décollement levels arises from variations in lithology (i.e., layer-parallel décollements) or structural features (i.e., inherited faults) that often promote localization of deformation as a result of increased stress contrasts and lower strength thresholds (Vogt et al., 2017; Broderie et al., 2018). The lithology of a décollement layer (salt, anhydrite, gypsum or shale), its stratigraphic position in the rock sequence (basal or middle), its thickness, and its fluid content

control the distribution and geometry of faulting and folding during crustal convergence (Simpson, 2009; Ruh et al., 2012; Najafi et al., 2014; Santolaria et al, 2022). Décollement layers with a certain thickness may lead to the formation of single-layer detachment folds (Mitra, 2003; Wallace and Homza, 1998). Furthermore, multiple décollement horizons within a shortened sedimentary sequence may connect along thrust faults and form structural ramp-flat geometries (Boyer and Elliott, 1982; Dal Zilio, 2020).

Besides weak layers within the stratigraphy, pre-existing basement faults, inherited from older tectonic events, can influence the location and geometry of new faults and folds and their propagation and termination in the overlying sedimentary cover, by serving as zones of weakness during compressional deformation. This ultimately affects the overall structural architecture of a fold-and-thrust belt (Bonini, 2012; Granado and Ruh, 2019; Parizot et al., 2022). The effects of fault inheritance can be complex and dependent on various factors, such as the orientation, dip, geometry, and depth of the inherited structures (White et al., 1986), the timing and style of subsequent deformation

(Zwaan et al., 2022), and the mechanical properties of the rocks involved (Ruh and Vergés, 2018).

The Fars Arc, located in the southeastern part of the Zagros Orogen (Fig. 1), offers an ideal area to study the combined influence of cover mechanical stratigraphy and basement inheritance. It is a typical example of a fold-and-thrust belt that contains multiple mechanically weak stratigraphic layers in the Phanerozoic sedimentary cover, as well as pre-existing inherited faults in the Precambrian crustal basement (Jackson and Fitch, 1981; Berberian, 1995; Talebian and

Jackson, 2004; Mouthereau et al., 2006; Yamato et al., 2011; Karasözen et al., 2019; Najafi et al., 2021). According to seismic, well-logging, and surface geological data, the Ediacaran–Lower Cambrian Hormuz series constitutes the main décollement in the Fars Arc. It consists of ~2 km of salt-bearing evaporites with minor carbonate and shale layers and decouples the crustal basement from the overlying sedimentary cover (Kent, 1958; Sherkati et al., 2005; Callot et

al., 2007; Leturmy et al., 2010). The presence of a thick salt layer at the base of the sedimentary cover in the Fars Arc is responsible for short-wavelength folds (Mouthereau et al., 2006). In some places, its basal décollement is spatially interrupted by basement faults related to the Cenozoic convergence (Kent, 1958; Talbot and Alavi, 1996; Sepehr and Cosgrove, 2004; Sherkati et al., 2005; Callot et al., 2007). The recent activity of these faults in the Fars Arc is documented by seismological data (Jackson and Fitch, 1981; Berberian and King, 1981; Berberian, 1995, Talebian

and Jackson, 2004; Tatar and Hatzfeld, 2004; Karasözen et al., 2019; Nissen et al., 2019). Focal mechanisms of seismic

events in the basement exhibit rather steep fault dip angles, suggesting the reactivation of inherited normal faults (Jackson, 1980).

In the Fars Arc, the activity of inherited faults has influenced the progression of deformation towards the foreland, with the Mountain Front Fault being associated with basement thrusting (Bahroudi and Koyi, 2003; Mouthereau et al., 2006, 2007a; Yamato et al., 2011; Ruh et al., 2014; Najafi et al., 2021). According to Mouthereau et al. (2006), basement deformation and thickening play a critical role in the Zagros Folded Belt, helping to explain the observed topographic growth. They also emphasize that the reactivation of pre-existing faults during the early stages of compression in the Zagros foredeep suggests a significant influence of inherited structural features on present-day deformation. Balanced cross-sections support this by demonstrating that basement involvement is necessary to account for varying base topographic elevations in Paleozoic and Mesozoic formations (Blanc et al., 2003; Molinaro et al., 2005; Mouthereau et al., 2007a). However, other studies propose that the most substantial impact of basement deformation on surface structures occurred later in the region's tectonic history, particularly during the Pliocene and Pleistocene (Molinaro et al., 2005; Sherkati et al., 2005; Tavani et al., 2018; Vergés et al., 2011; Najafi et al., 2018; Etemed-Saeed et al., 2020). It is widely accepted that the high-angle reverse faults initially formed as normal faults in the Arabian basement during the Permian–Triassic rifting of the Neo-Tethys Ocean (Navabpour et al., 2010), remaining inactive through the Jurassic to Late Cretaceous passive margin phase, before being reactivated during the Cenozoic collision between the Arabian and Eurasian plates.

Various 2D numerical modeling studies have investigated the evolution of fold-and-thrust belts and salt-bearing basins. Nilforoushan et al. (2013) demonstrated the influence of geothermal gradients and basement mineralogy on fault geometry and basement reactivation in the Fars arc, emphasizing the role of weak salt horizons in mechanical decoupling. Heydarzadeh et al. (2020) analysed factors such as sedimentation rates, erosion, and salt layer properties in Dehdasht basin, highlighting the importance of balanced surface processes and deformation rates. Humair et al. (2020) conducted simulations to study the interaction of folding and thrusting during Swiss Jura and Canadian Foothills fold-and-thrust belt evolution, focusing on the effects of layer-parallel shortening and initial geometrical perturbations. Their work showed that the magnitude of these perturbations influences whether folding or thrusting predominates, affecting the structural evolution and asymmetry of anticlines. Spitz et al. (2020) conducted 3D thermo-mechanical numerical simulations to investigate the influence of laterally variable inherited structures on fold-and-thrust belt evolution and nappe formation on Helvetic nappe system. The study demonstrated the fundamental importance of tectonic inheritance on fold-and-thrust belt evolution, with strain localization, folding, and nappe transport controlled by initial geometrical and mechanical heterogeneities. Almost all studies have focused on examining the collisional phase and deformation resulting from compression in the fold belts and the Fars Arc, while the earlier extensional history and its effect on later deformation have received less attention (e.g. Granado and Ruh, 2019). Incorporating a rifting phase into the model setup result in more realistic initial conditions for the convergence stage, featuring variations in crustal thickness, rift-related sedimentary basins and the presence of weak zones.

The interaction of the basal décollement level and the pre-existing basement faults and their combined influence on the distribution of thin- and thick-skinned tectonic styles in the Fars Arc, during both extensional and compressional phases, is not fully understood yet. Understanding these processes is crucial for improving our geological models of

the region, which has significant implications for hydrocarbon exploration. Although there is a general consensus on the importance of the décollement layer and basement faults, the detailed dynamics and their broader impact on regional tectonics require further investigation in during inversion tectonics.

The goal of our study is to test the impact of the Hormuz salt, and the mechanical properties and geometries of the basement faults inherited from the Permian continental rifting on the structural evolution of the Fars Arc. To achieve this goal, we conducted a series of two-dimensional thermo-mechanical numerical experiments to investigate how the presence or absence of fault inheritance (planar or listric) controls the deformation during rifting and subsequent tectonic shortening. Furthermore, we test the effect of a weak intermediate salt décollement of variable rheology on the structural evolution during convergence.

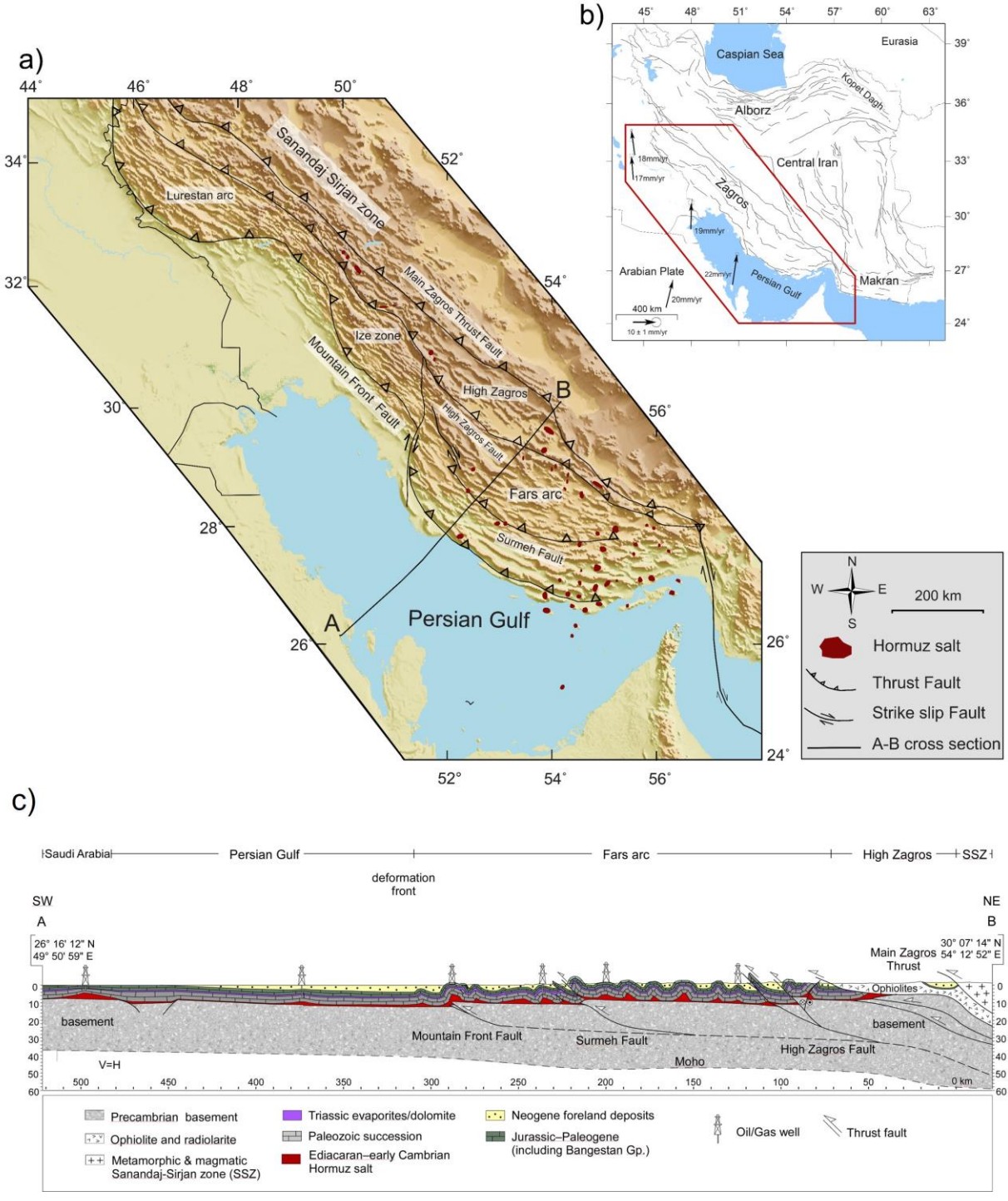

**Figure 1: (a) Overview of the geotectonic situation of Iran. (b) Main tectonic features in the Zagros fold-and-thrust belt, located along the northeastern margin of the Arabian Plate. (c) Regional geological cross-section, showing the crustal geometry (modified from Etemad-Saeed et al. 2020; Najafi et al. 2021; Najafi and Lajmorak 2020).**

120

## 2 Geotectonic and geological setting

The Zagros fold-and-thrust belt (ZFTB) is a NW-SE-trending orogenic belt that resulted from the convergence and continental collision between Arabian and Eurasian plates (Agard et al., 2011; Mouthereau et al. 2012; Vergés et al., 2024). The ZFTB extends over approximately 2000 km from the Taurus Mountains in Turkey in the northwest, to the Makran accretionary wedge in southeastern Iran (Fig. 1). Across strike, the ZFTB is bounded by the Main Zagros Thrust (i.e., the inherited suture of the old subduction zone) in the northeast, and by the Zagros Frontal Fault system in the southwest (Fig. 1b).

The NE margin of the Arabian Plate has been shaped by a series of tectonic events that ultimately resulted in the emergence of the ZFTB (Agard et al., 2011; Vergés et al., 2024; Madanipour et al., 2024): 1) Permian–Triassic rifting and opening of the NW-SE-trending Neo-Tethys Ocean, 2) Jurassic–Late Cretaceous passive marginal stage, 3) Late Cretaceous ophiolite obduction, 4) Oligocene soft continental collision between Arabian and Eurasian plates, and 5) middle Miocene-to-recent folding propagation across the mountain range.

The Permian–Triassic opening of the Neo-Tethys Ocean separated the Arabian Plate to the southwest from the Iranian microplate to the northeast (Szabo and Kheradpir, 1978; Berberian and King, 1981; Agrad et al., 2005). Lithospheric extension related to continental rifting deformed the Arabian crystalline basement and produced a series of NW-trending half-grabens parallel to the current orientation of the Zagros orogen (e.g. Jackson and Fitch, 1981; Mouthereau et al., 2007a). Following the rifting episode, the Arabian margin became passive during the Jurassic and Cretaceous (e.g. Alavi, 2004; Agard et al., 2005). An early stage of contractional deformation occurred in the Late Cretaceous with the obduction of the Neo-Tethys ophiolite and radiolarite slices onto the Arabian margin, presently preserved in Kermanshah, Neyriz, and Hajiabad (Agard et al., 2005; Saura et al., 2011; Bernaola et al., 2011).

From the Oligocene onward, continental collision led to the formation of the ZFTB, age-constrained by recent thermo-chronometric and magnetostratigraphic data (Pirouz et al., 2017; Koshnaw et al., 2019; Barber et al., 2019). The main phase of folding in the Zagros took place during the Miocene to Pleistocene, and progressively propagated to the SW (Hessami et al., 2001; Khadivi et al., 2009; Mouthereau et al., 2012; Ruh et al., 2014; Vergés et al., 2019; Najafi et al., 2021). GPS measurements indicate that the present-day convergence rate is approximately 20–30 mm/yr in a roughly N-S direction, of which ~6.5 ± 2 mm/yr is being consumed across the Zagros (Nilforoushan et al., 2003; Walpersdorf et al., 2006).

Based on its structural and stratigraphic characterizations, the Zagros is divided into the High Zagros imbricated zone in the NE, and the Zagros Simply Folded Belt (ZSFB) in the SW, separated by the High Zagros Fault (Fig. 1). The ZSFB extends to the Persian Gulf and the present-day Mesopotamian foredeep basin. It represents the deformed foreland of the orogeny, and displays elongated folds of regular wavelength (Falcon, 1974; Sepehr and Cosgrove, 2004; Mouthereau et al., 2006). The ZSFB consists of an 8–14 km-thick sedimentary sequence covering metamorphosed rocks of the Precambrian Arabian basement (Mouthereau et al., 2007b; Lacombe et al., 2011). Based on lateral stratigraphic and structural variations, the ZSFB is divided, from NW to SE, into the Lurestan Arc, the Izeh zone, the Dezful Embayment, and the Fars Arc (Fig. 1).

In this study, we focus on the Fars Arc, the largest tectonic domain of the Simply Folded Belt, limited by the Kazerun–Borazjan segmented dextral fault system to the west and the Minab–Zendan–Palami fault system to the east (Fig. 1b;

Regard et al., 2004; Lacombe et al., 2011). The Fars Arc extends over more than 300 km across strike, is ~200 km wide from the High Zagros Fault to the deformation front and developed as a result of folding of the thick sedimentary

cover (Stöcklin, 1974, Berberian and King, 1981). Its elongated folds show a distinctive periodic pattern with axial lengths reaching to 200 km and wavelengths of ~15 km (Fig. 1c; Mouthereau et al., 2007b; Najafi et al., 2021). The Fars Arc is scattered with salt diapirs, a majority of them located in the eastern part of the arc (Sherkati et al., 2006; Callot et al., 2007).

The stratigraphy of the Fars Arc contains a major basal and several minor intermediate mechanically weak

décollement layers (Fig. 2). Near the base of the sedimentary cover, the ~2 km-thick salt-bearing evaporites of Hormuz Formation of Ediacaran–Early Cambrian age overlay early Ediacaran sediments (Callot et al., 2007). The 3-km-thick Paleozoic sequence is dominantly composed of sandstone, dolomite, and shale. The Dehram Group with a thickness of more than 1 km was deposited during the Permian–Triassic. The Triassic Dashtak Formation of 550–850 m thickness overlies the Dehram Group and plays as the minor intermediate décollement level in the frontal Fars Arc

(Motamedi et al., 2012; Najafi et al., 2014). It laterally grades into the dolomites of the Khanehkat Formation in the interior of the range, where it loses its efficiency as a décollement level (Szabo and Kheradpir, 1978). During the passive marginal stage in the Jurassic–Late Cretaceous, the Zagros basin was characterized by a shallow-marine environment, when the 2-km-thick carbonate and minor detrital and evaporite successions of the Khami and Bangestan groups were deposited (Sharp et al., 2010).

The Neo-Tethyan oceanic crust was obducted over the Arabian plate margin during the Late Cretaceous, and loading resulted in the flexure of the Arabian lithosphere. This produced an early foreland basin in the Lurestan region of the NW Zagros, referred to as the Amiran foreland basin by Homke et al. (2009) and Saura et al. (2011). The thick successions of deep-water shales of the Paleocene to Eocene formations indicate regional subsidence. The passive margin succession of the Paleocene to Miocene sediments has a thickness of ~3 km (Jahani et al., 2009; Najafi et al.,

2014). In the Oligocene, the Asmari Formation was deposited over the Pabdeh Formation in the Fars region. The post-Asmari clastics, known as the Fars Group, including the Gachsaran, Mishan, and Aghajari formations (early Miocene to Pliocene), have a total thickness of about 3 km. Migration of the Aghajari-Bakhtyari sedimentary system towards the foreland and the propagation of folding were in sequence. They have migrated at a rate of 20 mm/yr in the Fars Arc and 15 mm/yr in the Lurestan Arc during the last 20 Myr (Ruh et al., 2014; Vergés et al., 2019).

In the Fars Arc, two main décollement levels occur in the stratigraphy (Fig. 2); the Hormuz Formation represents the basal and the major décollement level, and the evaporites of the Dashtak Formation constitute a minor intermediate décollement level (Callot et al., 2012; Motamedi et al., 2012; Najafi et al., 2014).

Seismic activity at mid-crustal depths provides key evidence for basement involvement in the Zagros. Most earthquake centroid depths range from 4 to 25 km, affecting both the basement and cover, with many exhibiting reverse focal

mechanisms (Jackson and Fitch, 1981; Berberian, 1995; Talebian and Jackson, 2004; Karasözen et al., 2019). In the Fars Arc, the major inherited basement reverse faults, from SW to NE, include the Mountain Front Fault, the Surmeh Fault, the High Zagros Fault, and the Main Zagros Thrust (Fig. 1c). A geological cross-section of the Fars Arc reveals evidence of both thin-skinned and thick-skinned tectonic deformation occurring simultaneously (Mouthereau et al.,

2007b; Najafi et al., 2021; Fig. 1c). This deformation is expressed through large-scale detachment folds and forced folds (Jackson, 1980; Lacombe et al., 2011; Mouthereau et al., 2012).

While numerous studies have focused on the Zagros fold-and-thrust belts and the Fars Arc, several crucial aspects of the Fars Arc's geological evolution remain poorly understood. Specifically, the exact mechanisms and timing of basement involvement, the interaction between basement faults and salt décollements during tectonic inversion, and the relative influence of thin-skinned versus thick-skinned tectonics on the overall structural evolution are still unresolved (Mouthereau et al., 2006; 2012). To address these uncertainties, we employ a numerical model that simulates the full tectonic history of the Fars Arc, including both an initial extensional phase and a subsequent compressional phase.

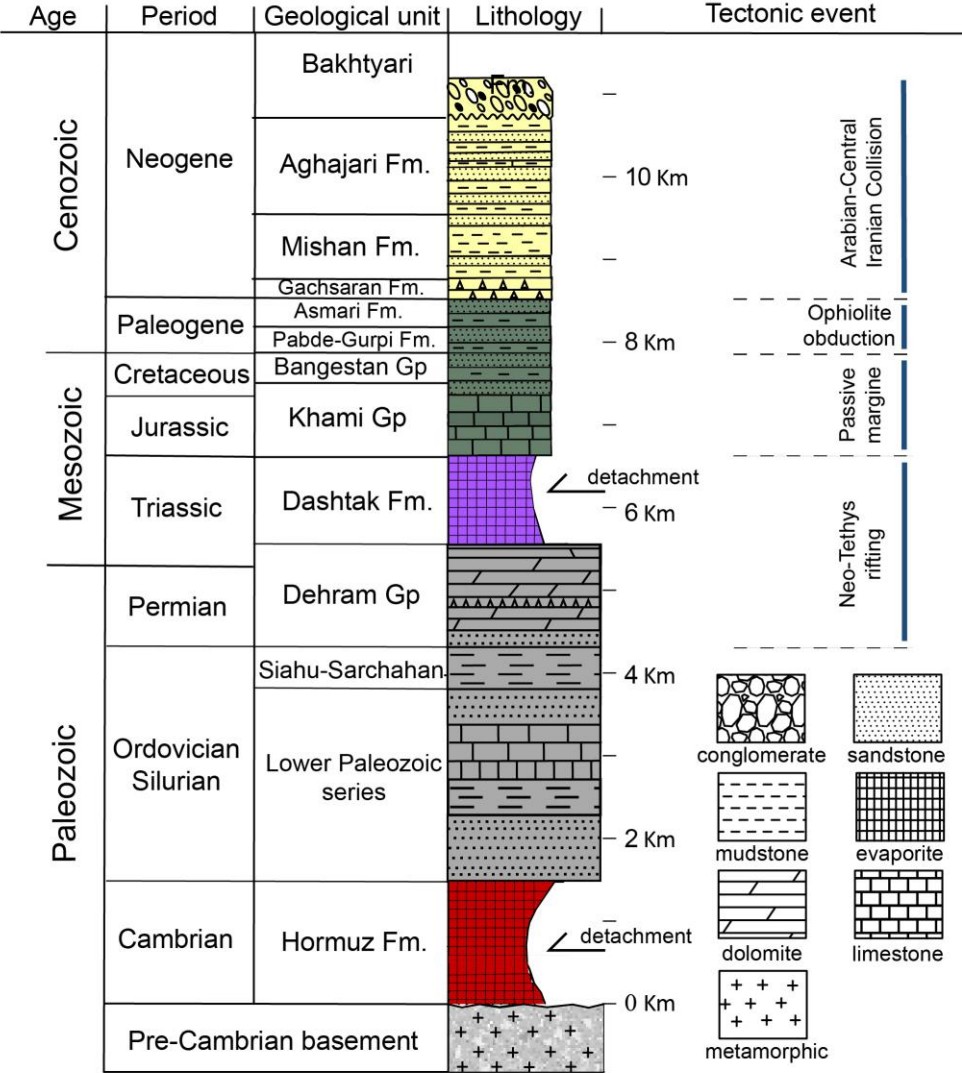

Figure 2: Stratigraphic column of the sedimentary cover in the Fars Arc (modified from Callot et al., 2007; Jahani et al. 2009; Motamedi et al., 2012; Mouthereau et al. 2007; Najafi et al. 2014; Sepehr and Cosgrove 2004; Sherkati et al. 2006). Violet and red lithologies represent regional décollement horizons.

## 3 Numerical Model

In order to investigate the role of inherited extensional faults and mechanically weak décollement horizons on the structural evolution of the Fars Arc, a series of numerical experiments were conducted. We apply the two-dimensional finite difference numerical code "Norma" (Ruh et al., 2022), with a fully staggered Eulerian grid, freely moving Lagrangian markers (marker-in-cell technique; Gerya, 2019), and a temperature-dependent visco-elasto-plastic/brittle rheology.

### 3.1 Governing equations

The numerical code solves for the conservation of mass and momentum (the Stokes equation) on a Eulerian grid to calculate the velocity and pressure fields:

$$\frac{\partial v_i}{\partial x_i} = 0 \tag{1}$$

and

$$-\frac{\partial p}{\partial x_i} + \frac{\partial \tau_{ij}}{\partial x_j} = \rho g_i, \tag{2}$$

where $v_i$ and $x_i$ are velocities and spatial coordinates, $p$ the dynamic pressure (mean stress), $\tau_{ij}$ the deviatoric stresses, $\rho$ the density (constant), and $g_i$ the gravitational acceleration ($g_1 = 0$; $g_2 = 9.81 \frac{m}{s^2}$),. Temperature is solved by considering the energy conservation on the Eulerian grid:

$$\rho C_p \left( \frac{DT}{Dt} \right) = \kappa \frac{\partial^2 T}{\partial x_i^2}, \tag{3}$$

where $C_p$ is isobaric heat capacity, $\frac{D}{Dt}$ is the total time derivative, $T$ is temperature, $t$ is time, $\kappa$ is the thermal conductivity coefficient. Additional heat production, such as radioactive heating and shear heating, is not activated in the presented experiments due to the geometrical constraints of the model setup and related boundary conditions that affect the diffusion of such secondary heat production.

The pressure and temperature fields calculated from the governing equations are interpolated onto the Lagrangian markers based on a linear distance-weighted scheme (Gerya, 2019). The Lagrangian markers store all material properties and advect through the Eulerian grid based on a fourth-order Runge-Kutta interpolation of the calculated two-directional Eulerian velocity field.

### 3.2 Rheological model

The numerical code employs a visco-elasto-plastic/brittle rheology, where visco-elasticity is implemented by a Maxwell-type expression. Elasticity plays a key role in capturing short-term stress accumulation and release, crucial for fault and fold behaviour. The Maxwell model allows the simulation of both immediate elastic response and long-term viscous flow, ensuring that important transient phenomena such as fault reactivation and seismic activity are

accurately represented during both extension and convergence. The strain rate of a Maxwell body under stress consists of viscous and elastic components:

$$\dot{\varepsilon}_{ij} = \frac{1}{2\eta}\tau_{ij} + \frac{1}{2G}\frac{D\tau_{ij}}{Dt}, \tag{4}$$

where $G$ is the elastic shear (100 GPa for all materials here)and $\eta$ is the effective viscosity. First-order finite differences in time are used to represent the objective co-rotational time derivatives of the visco-elastic stresses:

$$\frac{D\tau_{ij}}{Dt} = \frac{\tau_{ij}-\tau_{ij}^{old}}{\Delta t}. \tag{5}$$

The effective viscosity $\eta$ is calculated from the non-Newtonian dislocation creep equation:

$$\eta_{disl} = 0.5 \cdot \frac{1}{A_D} \cdot \sigma_{II}^{(1-n)} \cdot \exp\left(\frac{Q}{RT}\right), \tag{6}$$

where

$$\sigma_{II} = \sqrt{\frac{1}{2}\tau_{ij}^2}, \tag{7}$$

and $R, A_D, n,$ and $Q$ are the gas constant, the pre-exponential factor, the power-law constant, and the thermal activation energy, respectively. Elastic behavior is achieved by updating the effective viscosity in response to an elastic time step (Maxwell time; $\Delta t_e = 1000$ years) and the stress history (Gerya and Yuen, 2007; Moresi et al., 2007; Moresi et al., 2003). The updated visco-elastic deviatoric stresses are defined by:

$$\tau_{ij} = 2\eta\dot{\varepsilon}_{ij}z + \tau_{ij}^{old}(1-Z), \tag{8}$$

where

$$Z = \frac{\Delta t_e \cdot G}{\eta_{disl}+\Delta t_e \cdot G}. \tag{9}$$

The effective viscosity is multiplied by this visco-elastic factor ($Z$) to obtain a numerical viscosity $\eta_{num}$ to be used for solving the set of equations:

$$\eta_{num} = \eta_{disl} \cdot Z = \frac{\eta_{disl} \cdot \Delta t_e \cdot G}{\eta_{disl}+\Delta t_e \cdot G}. \tag{10}$$

Plastic failure occurs if the second invariant of the visco-elastic stress tensor exceeds the yield stress, $\sigma_y$, following a pressure-dependent Drucker-Prager criterion:

$$\sigma_y = p \cdot (1-\lambda) \cdot sin\varphi + c \cdot cos\varphi, \tag{11}$$

where $p$ is the pressure (mean stress), and $c, \varphi$, and $\lambda$, are the cohesion, friction angle, and the fluid pressure ratio of the bulk material, respectively. The components of stress ($\sigma_{xx}$ is the normal stress component, and $\sigma_{xy}$ is the shear stress component) and the viscosity are then updated as:

$$\sigma_{xx}^{new} = \begin{cases} \sigma_{xx}\frac{\sigma_y}{\sigma_{II}}, & \text{if } \sigma_{II} > \sigma_{yield} \\ \sigma_{xx}, & \text{if } \sigma_{II} \leq \sigma_{yield} \end{cases}, \tag{12}$$

$$\sigma_{xy}^{new} = \begin{cases} \sigma_{xy}\frac{\sigma_y}{\sigma_{II}}, & \text{if } \sigma_{II} > \sigma_{yield} \\ \sigma_{xy}, & \text{if } \sigma_{II} \leq \sigma_{yield} \end{cases}, \tag{13}$$

$$\eta_{num} = \frac{\sigma_y}{2\dot{\varepsilon}_{II}}, \quad \text{if } \sigma_{II} > \sigma_{yield}, \tag{14}$$

where

$$\dot{\varepsilon}_{II} = \sqrt{\frac{1}{2}\dot{\varepsilon}_{ij}^2}. \tag{15}$$

The viscosities, including the visco-elastic effect and the plastic failure, are calculated on the Lagrangian markers and interpolated onto the Eulerian nodes using a distance-weighted scheme. To ensure numerical stability, they are capped by lower and upper cutoffs of $10^{17}$ and $10^{25}$ Pa·s, respectively. Given the numerical time step of $\Delta t_e = 1000$ years, material undergoing deformation at viscosities below $3.16 \times 10^{21}$ Pa·s can be considered predominantly viscous, while deformation at viscosities above $3.16 \times 10^{21}$ Pa·s is to a significant part elastic and thus reversible. Elastic relaxation

time varies between ~1 year and 1 million years, depending on the viscosity of the material, which results in Deborah numbers of $10^{-7} – 0.1$ for a deformation period of 10 million years.

### 3.3 Initial geometrical setup

The model domain is defined by a box of 500 km width and 60 km height (Fig. 3a). The Eulerian grid consists of 1001 $\times$ 121 nodes, with a nodal resolution of $500 \times 500$ m. The initial marker distribution defines, from bottom up, 1) a 30-

km-thick crustal basement layer, given the depth of basement crustal detachment (Vergés et al., 2011; Kendall et al., 2019), 2) a mechanically-weak salt horizon emplaced between 100 km < x < 450 km, 3) a 3-km-thick Palaeozoic sequence based on the stratigraphy of the Fars Arc, and 4) a 25-km-thick low-density and low-viscosity sticky-air layer to simulate a free surface, allowing for the vertical growth of the evolving fold-and-thrust belt (Crameri et al., 2012). The thickness of the salt layer varies in different models in order to examine its impact on the structural

evolution. Some experiments include three inherited basement faults located at x = 100, 200, and 350 km (top of the basement), dipping at an angle of 50° at their top towards the hinterland (Mouthereau et al., 2006). These weak zones may either have a listric or planar geometry. The rheological parameters used in the experiments are provided in Table 1. Strain weakening is implemented by a linear decrease of the values for frictional angle ($\varphi$) and cohesion ($c$) between a lower and upper strain threshold, defined by the second invariant of the strain tensor $\varepsilon_{II}^0 = 0.1$ and $\varepsilon_{II}^1 = 1$.

The initial temperature field is characterized by a linear temperature increase with depth, starting from 0 °C at the interface between rock and sticky-air and reaching 600 °C at the bottom, in agreement with a constant geothermal gradient of 20 °/km inferred from paleothermal data (Aldega et al., 2018) and a relatively thick lithosphere (~200 km) beneath the Zagros (Jiménez-Munt et al. 2012; Priestley et al. 2012; Tunini et al., 2014). Applying a linear geotherm is a reasonable simplification for the uppermost ~50 km of the continental lithosphere (Hasterok and Chapman, 2011;

Goes et al., 2020). Each Eulerian cell initially contains 16 randomly distributed Lagrangian markers carrying rock information and properties. If the finite spatial domains of a specific node become empty of any Lagrangian marker, the previous interpolated parameters are applied for solving the system of equation of this particular node.

Table 1: Applied rheological parameter

| Rock Type | | $A_D(Pa^{-n})$ | $Q(KJ\ mol^{-1})$ | $n$(-) | $\rho(kgm^{-3})$ | $\varphi(°)$† | $c(MPa)$† | $\lambda$(-) | $k(Wm^{-1}K^{-1})$ | $C_p(m^2K^{-1}s^{-2})$ |
|---|---|---|---|---|---|---|---|---|---|---|
| Sticky-air | | $10^{-17}$ | - | 1 | 1 | - | - | - | 200 | $3\times10^6$ |
| Sediments [a] | | $5\times10^{-18}$ | 154 | 2.3 | 2500 | 10 | 1 (0.1) | 0.4 | 2.5 | $1\times10^3$ |
| Syn-rift sediments [a] | | $5\times10^{-18}$ | 154 | 2.3 | 2500 | 30 (20) | 1 (0.1) | 0.4 | 2.5 | $1\times10^3$ |
| Paleozoic series [a] | | $5\times10^{-18}$ | 154 | 2.3 | 2700 | 30 (20) | 1 (0.1) | 0.4 | 2.5 | $1\times10^3$ |
| Salt layer | Non-linear [b] | $1.82\times10^{-39}$ | 32.4 | 5 | 2200 | - | - | - | 2.5 | $1\times10^3$ |
| | Linear | $10^{-18}$ | - | 1 | | | | | | |
| Basement rock [c] | | $6.31\times10^{-20}$ | 276 | 3.05 | 2800 | 30 (20) | 10 (0.1) | 0.4 | 2.5 | $1\times10^3$ |
| Basement fault [c] | | $6.31\times10^{-20}$ | 276 | 3.05 | 2800 | 10 | 0.1 | 0.4 | 2.5 | $1\times10^3$ |

[a] **Quartzite (**Ranalli and Murphy, 1987; Stöckhert et al., 1999)
[b] **Rock salt (**Li and Urai**,** 2012)
[c] **Diabase (**Wilks and Carter, 1990**)**
† Values in brackets indicate strain weakened value.

### 3.4 Boundary conditions

The numerical experiments simulate the deformation of the NE margin of the Arabian Plate during Permian–Triassic rifting and Oligocene–recent continental collision. Tectonic quiescence affects the thermal state and, consequently, the rheological behaviour of the rocks. However, we assume that the temperature field reached a steady state by the onset of the collision phase to simplify the model setup. This implementation allows us to effectively capture the relevant thermal conditions while maintaining focus on the critical dynamics of the extension and collision phases.

The velocity boundary conditions are prescribed in a way to simulate the tectonic inversion of the rifted margin, including initial extension and subsequent convergence. During the rifting phase, an outward horizontal boundary velocity of $v_x$ = 5 mm/yr is applied on the right side of the model domain, while the left boundary is kept fixed horizontally (Fig. 3a). The bottom boundary has zero vertical velocity, and the top boundary has an incoming vertical velocity of $v_y$ = 0.6 mm/yr, ensuring the conservation of volume within the model domain (Fig. 3a). On all boundaries

free-slip conditions (zero shear stress) are prescribed. The finite Eulerian domain impedes elastic bending of the lower boundary, affecting resulting surface taper angles compared to the ZFTB (Mouthereau et al., 2006; McQuarrie, 2004). The extension phase is applied for a period of 5 Myr, resulting in a total extension of 25 km. During this period the rift basin is allowed to be filled with syn-rift deposits (equivalents of the Dashtak and Dehram formations). Following the extension phase, 4 km of sediments representing the depositional environment during the tectonic quiescence and

subsidence period are added onto the existing stratigraphy (Fig. 3b). These sediments represent the Khami and Bangestan groups, and the Gurpi and Pabdeh formations (Fig. 2). The Late Cretaceous ophiolite obduction episode has affected only some parts of the Arabian margin, namely the Kermanshah, Neyriz, Hajiabad and Oman regions. We have not included the Late Cretaceous deformation in our modeling, as it accounts for just a few percent of the observed shortening in the ZSFB (e.g. Saura et al., 2011).

Crustal shortening in the experiments is simulated by imposing a horizontal velocity of 1 cm/yr at the bottom and the left side of the domain. The right-side acts as a partial backstop by allowing the basement part of the crust to escape the model but preventing the sedimentary cover from exiting. For the lower part of the right boundary (basement), different horizontal velocities between 0 and 1 cm/yr are tested (Fig. 3b). This arrangement permits us to vary the degree of basement involvement and to study the development of thin-skinned and thick-skinned tectonics in the

model. The compressional phase ran for 15 Myr, accommodating a total shortening of 25% (i.e., 125 km), This amount offsets the 25 km extension during the extensional phase and the 20% shortening during the collisional phase, comparable to estimates for the SE Zagros (McQuarrie, 2004; Motamedi et al., 2012; Pirouz et al., 2017; Najafi et al., 2021).

### 3.5. Surface processes

The surface processes of sedimentation and erosion during the evolution of the models are simulated by applying diffusion of the rock/air interface. We used the diffusion equation:

$$\frac{\partial h_s}{\partial t} = K \frac{\partial^2 h_s}{\partial x^2}, \tag{16}$$

where $h_s$ is the surface topography, $t$ is time, $K$ denotes a diffusion constant ($K = 10^{-6}\ \frac{m^2}{s}$), and $x$ is the horizontal

coordinate. The left and right sides of the surface line prescribe free-slip boundaries.

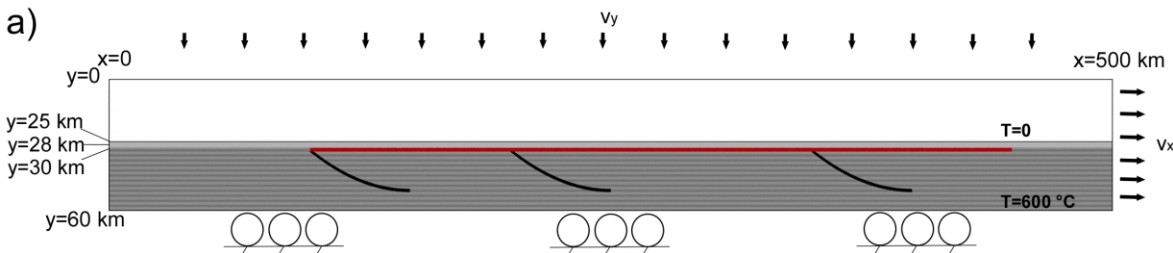

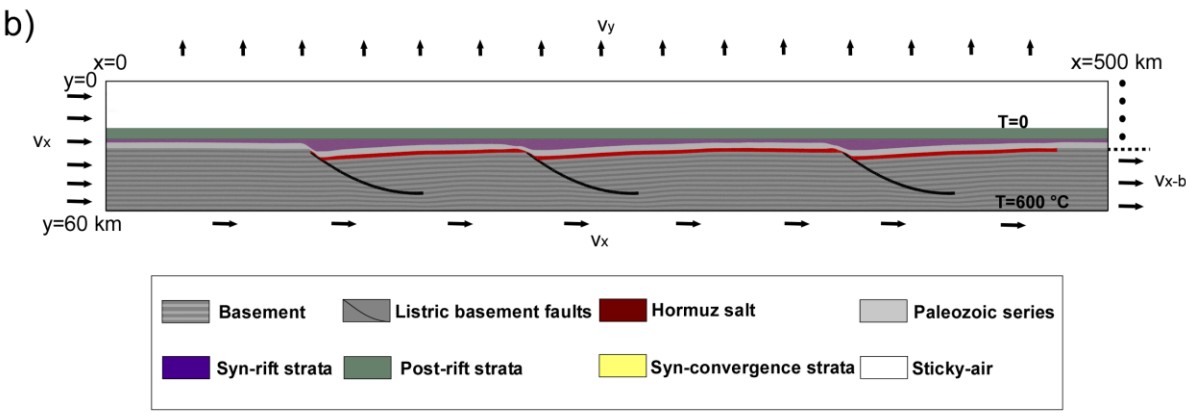

**Figure 3: Setup and boundary conditions of numerical modeling. (a) Initial compositional setup for extension phase with listric faults. (b) Fully extended model (5 Myr) with listric faults before convergence. $v_x$, $v_y$ and $v_{x\_b}$ is horizontal velocity, vertical velocity, and basement horizontal velocity, respectively.**

**4. Results of modeling**

A total of 11 experiments are presented, investigating the role of basement involvement and the impact of salt rheology and thickness on the structural evolution of the Fars Arc. Experiments are divided into four series varying in 1) the thickness of the salt layer, 2) the rheology of the salt layer, 3) the existence and geometry of pre-existing basement faults, and 4) the degree involvement of the basement during convergence (Table 2). While the numerical model is

based on a viscous formulation and thus develops localized shear zones, we refer to them as faults if they form due to the implemented Drucker-Prager failure criterion (see equation 11). First, we present the temporal evolution of the reference model, to which the rest of the models are compared. All experiments undergo 5 Myr of extension, followed by 15 Myr of compression. A link to videos (Graphics Interchange Format) of all numerical experiments can be found in Data Availability.


**Table 2: List of the numerical models**

| Model | Salt thickness (km) | Salt rheology | Fault geometry | Basement velocity (mm s$^{-1}$) | Figure |
|---|---|---|---|---|---|
| **Model 1** (Reference model) | 2 | Non-linear | listric | 0 | 4,5,8,11,12,14, S1 |
| **Model 2** | 0 | Non-linear | listric | 0 | 6,8,12, S2 |
| **Model 3** | 4 | Non-linear | listric | 0 | 6,8, 12,13, S3 |
| **Model 4** | 2 | Linear ($10^{18}$ Pa·s) | listric | 0 | 7,8,12, S4 |
| **Model 5** | 2 | Linear ($10^{20}$ Pa·s) | listric | 0 | 7,8 ,12, S5 |
| **Model 6** | 2 | Non-linear | planar | 0 | 9,12, S6 |
| **Model 7** | 2 | Non-linear | no fault | 0 | 9,12, S7 |
| **Model 8** | 2 | Non-linear | listric | 0.25 | 10,12, S8 |
| **Model 9** | 2 | Non-linear | listric | 0.5 | 10,12, S9 |
| **Model 10** | 2 | Non-linear | listric | 0.75 | 10,12, S10 |
| **Model 11** | 2 | Non-linear | listric | 1 | 10,12, S11 |

### 4.1 Evolution of the reference model

The reference model (Model 1) has an initial 2-km-thick salt horizon with a power-law viscous rheology (Table 1 and 2). The basement exhibits three inherited listric fault zones and is shortened at the same rate as the sedimentary cover during convergence. The extensional phase is characterized by the development of half-graben basins forming along the listric basement faults that are filled by syn-extensional deposits (Fig. 4a-c and S1). The second invariant of the strain-rate tensor shows that the fastest deformation occurs along the pre-existing weak zones. During extension, all three inherited faults are simultaneously active, without any recognizable preference. Normal fault deformation along the basement faults propagates upward into the post-salt strata inducing typical half-grabens and bending of the hanging wall (Fig. 4c and S1).

After 5 Myr of compression, the cover sequence mainly deforms between the right backstop and the salt pinch-out at x ≈ 150 km (Fig. 4d and S1). The basement fault closest to the backstop has experienced a significant amount of reverse motion and has caused a large step of several kilometres in the salt décollement geometry. Faulting of the sedimentary cover mainly develops where the syn-rift strata are the thinnest, i.e. in front of the basement thrusts. Strain localization occurs along the basement faults, and in the case of the frontal fault (x ≈ 120 km) it cuts across the strata overlying the salt layer (Fig. 4d and S1). Towards the backstop, the deformation in the basement and the cover sequence is decoupled along the weak salt horizon.

As convergence progresses (10 Myr), the basement fault at x ≈ 300 km undergoes inversion and creates a ramp-flat-ramp geometry (Fig. 4e and S1). Basement fault propagation deforms the syn-extensional strata and forms harpoon-like anticlines within the sedimentary cover. The syn-tectonic sedimentation concentrates within multiple structural basins between the developing anticlines. The salt layer decouples the deformation between the basement faulting and the sedimentary cover folding, allowing for more amplification within the sedimentary cover. Strain rates indicate intense deformation within the basement, which particularly localizes along the inverted normal faults, as well as the basal décollement level. The strain rate pattern starts to gradually diffuse below ~26 km depth, indicating the transition from brittle to ductile deformation in the lower crust (Fig. 4f and S1).

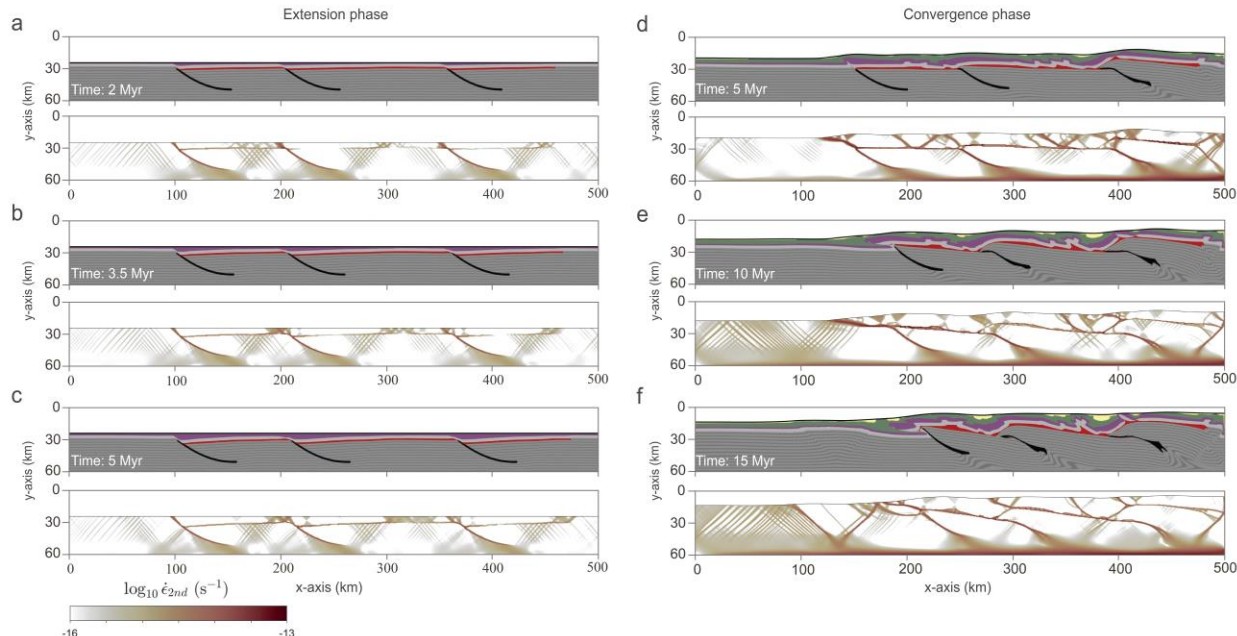

**Figure 4: Temporal evolution of the reference model during extensional (a-c) and compressional (d-f) phases. For each time panel, the compositional layers (top) derived from the Lagrangian markers, and the second invariant of the strain-rate tensor (bottom) are shown.**


Patterns of the second invariant of the stress tensor after the extensional phase and after full convergence indicate an increase in stress with depth down to ~30 km ($y = 50$ km), where the brittle-to-ductile transition begins (Fig. 5a,d). The lower part of the basement displays low stresses given its lower viscosities and ductile nature (Fig. 5b, e). Viscosity plots furthermore indicate the position of the low-viscous décollement and basement thrusts.

Figure 5c illustrates the vertically averaged viscosity of the non-Newtonian salt horizon after 25 km of extension, influenced by both temperature and strain rate (Table 1; Li and Urai, 2012). During extension and the formation of half-grabens along pre-existing faults, the strain rate and temperature increase in these areas. Consequently, these regions exhibit the lowest viscosities ($10^{18}$ Pa·s), corresponding to the locations of the basement faults (Fig. 5c). In areas without strain localization within the cover sequence, the viscosity shows larger values ($10^{19}$-$10^{20}$ Pa·s). After 5

Myr of extension, the half-graben basins are covered by post-rift deposits, prior to the onset of convergence. After 15 Myr, the intensity of thin-skinned folding within the sedimentary cover increases (Fig. 5f). The development of long-wavelength folds is influenced by the reactivation of basement faults, which exert a significant impact on the deformation of the upper crustal region. The frontal basement fault is influenced by the salt pinch-out and over-thrusts the post-rift strata, whereas the other two faults undergo a transition to ramp-flat-ramp configuration. In the post-rift

sediments, new shear zones are formed along the continuation of the pre-existing faults. These faults have been displaced due to the presence of basal salt, leading to deviation from their original dip. Strain rates indicate that the continuity of the salt décollement is disrupted due to the offset introduced by the inherited faults (Fig. 4f). The largest viscosities of salt are found in the anticlines with long wavelengths that form in the areas between two pre-existing

faults (Fig. 5f). In contrast, lower viscosities are observed in areas where the strain rate reaches its maximum,

specifically at the tip of the basement faults where the salt layer is thin.

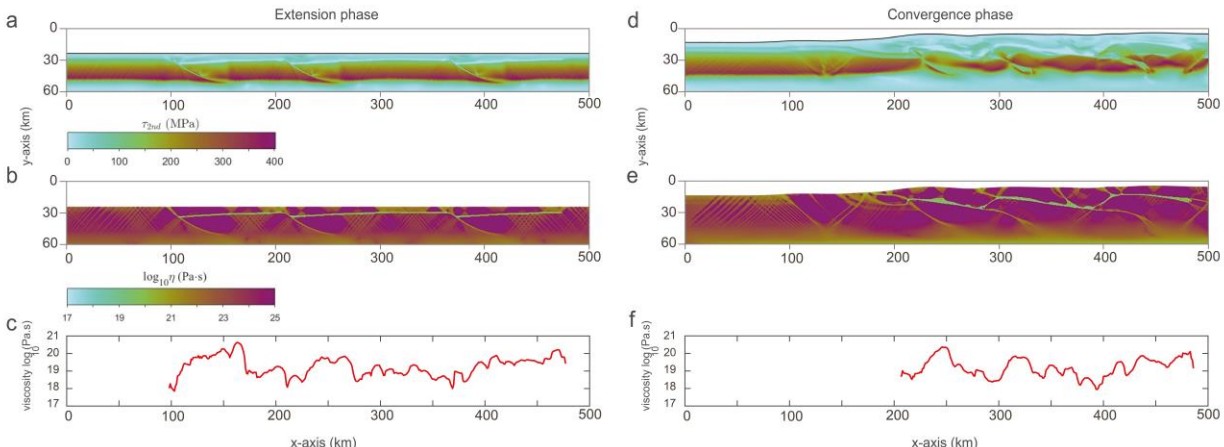

**Figure 5: Second invariant of the stress tensor and viscosity after full extension (a, b) and full convergence (d, e). c, f)**

**Vertically-averaged viscosity of the nonlinear salt horizon after 5 Myr of extension (c) and 15 Myr of convergence (f),**
**respectively.**

### 4.2 Effect of salt thickness

Figure 6 shows the effect of the thickness of the salt layer on the deformation. In the absence of a salt layer (Model 2; Fig. 6a), no mechanical decoupling occurs between the sedimentary cover and the basement. The extension phase is

dominated by the pre-existing faults and their geometry, and the majority of the deformation accumulates along the pre-existing normal faults. After 15 Myr of convergence, long-wavelength basement-cored anticlines develop in the sedimentary cover. The anticlines form asymmetrically and with low internal deformation of the sedimentary sequence (Fig. 6b and S2). The folding style within the sedimentary cover is determined by the pre-existing faults, resulting in the development of thrust faults that propagate upward through the overlying rocks. Basement faults cut the

sedimentary cover and reach the surface without any significant deviation from their initial orientation. Furthermore, backthrusts form in the hanging wall as conjugates to the reactivated inherited faults. Strain rate patterns reveal that the entire crustal package deforms uniformly, without mechanical decoupling along the different lithological layers. Increasing the salt layer thickness to 4 km (Model 3) does not significantly affect the structural development during the extension phase compared to the reference model (Fig. 4c and 6c). However, a thicker salt layer hinders the

extensional faults from fully cross-cutting the salt after full extension (Fig. 6c and S3). During shortening, a thicker salt layer acts as a more efficient décollement, impeding small-scale deformation in the sedimentary cover relative to the reference model (Fig. 6d; compare to Fig. 4f). The sedimentary cover develops décollement folds cored by a thickened salt. The cover is pushed over the frontal basement fault on the left, which upon reactivation results in a salt-cored fault-propagation fold at $x \approx 210$ km. Furthermore, the increased amount of salt allows for vertical

breakthroughs and diapir formation.

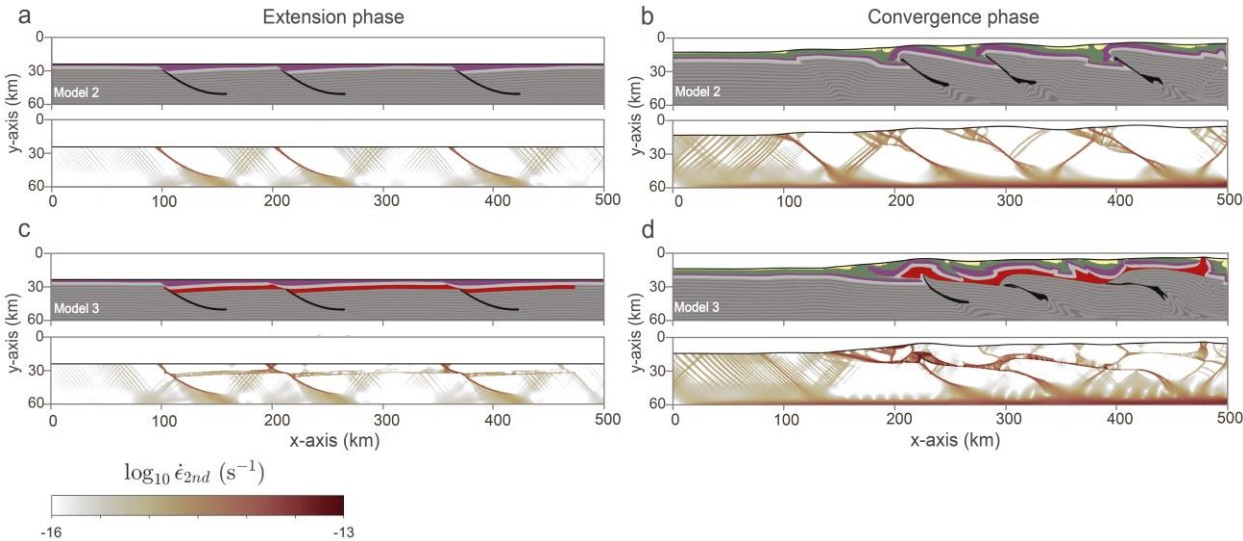

**Figure 6: Composition inferred from Lagrangian markers and the second invariant of the strain-rate tensor, affected by salt thickness after 5 Myr of extension and 15 Myr of convergence. (a-b) In the absence of salt layer. (c-d) In the presence of a salt layer with 4 km thickness.**

## 4.3 Effect of salt rheology

To investigate the impact of the rheology of the salt on the style of deformation, we considered two models with linear viscosity for the salt layer and compared the results with that of the reference model. For Models 4 and 5, we chose linear viscosities of $10^{18}$ and $10^{20}$ Pa·s, respectively, which are roughly equal to the lowest and largest viscosities observed in the reference model (Fig. 5c, f). After 5 Myr of extension, the low-viscosity Model 4 leads to the formation of symmetric ridges along the half-grabens and footwalls (Fig. 7a and S4). The majority of the deformation is accommodated by the salt layer and the inherited faults. After 15 Myr of compression, the pattern of folds within the sedimentary cover diverges from that of the reference model, as they display an inclination towards the hinterland (Fig. 7b and S4). Based on the strain rates, deformation is accommodated across the décollements.

In Model 5 with higher salt viscosity, the structural pattern after 5 Myr of extension resembles that of the reference model (Figs. 4c and 7c). Strain rates reveal significant deformation occurring within the basement, localizing along the pre-existing weak zones. After 15 Myr of shortening, the style of folding shows significant differences from Model 4 (Figs. 7b, d and S5). Several forethrusts develop above the imbricated basement blocks and backthrusts are generally absent. Strain rates indicate that the décollement remains laterally connected across the different basement blocks, whereas it is interrupted in the model with low linear viscosity (compare Fig. 7b and 7d).

Figure 8 presents the vertically-averaged strain rate within the salt layer across models with different décollement rheology. The influence of a power-law rheology on strain rate is characterized by a non-linear, accelerating response to stress. As stress levels increase, the strain rate increases more rapidly compared to a linear rheology, where the strain rate maintains a constant, linear relationship with stress. The mean strain rate in a salt layer with a power-law rheology exceeds that in a salt layer with linear rheology, as illustrated.

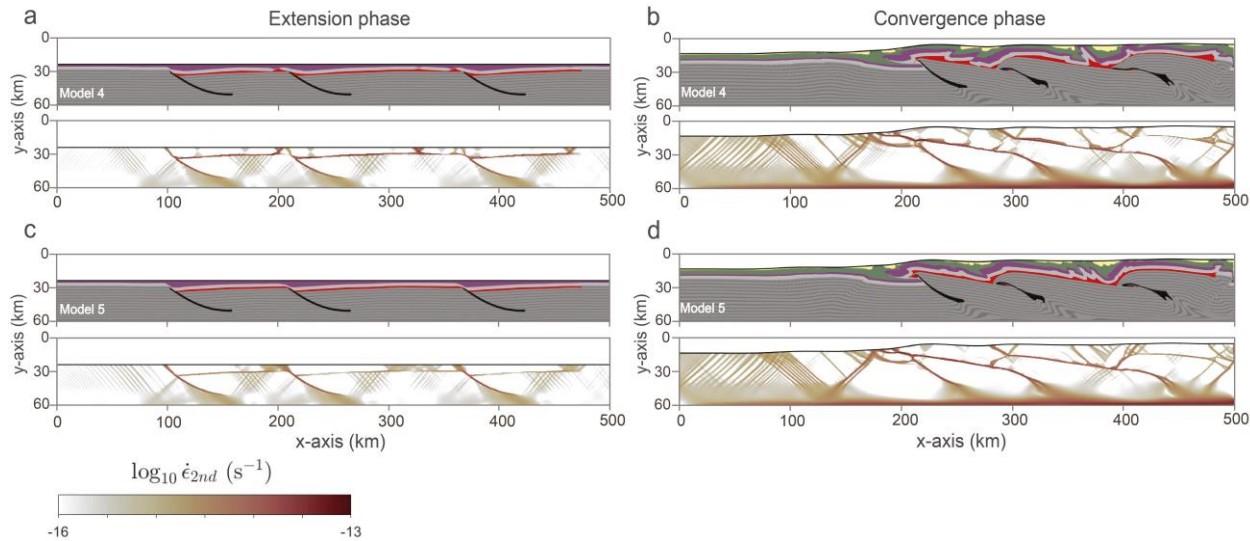

**Figure 7: Composition inferred from Lagrangian markers and the second invariant of the strain-rate tensor, affected by different viscosity for linear salt layer after 5 Myr of extension and 15 Myr of convergence. (a-b) linear salt horizon with $10^{18}$ Pa·s viscosity. (c-d) linear salt horizon with $10^{20}$ Pa·s viscosity.**

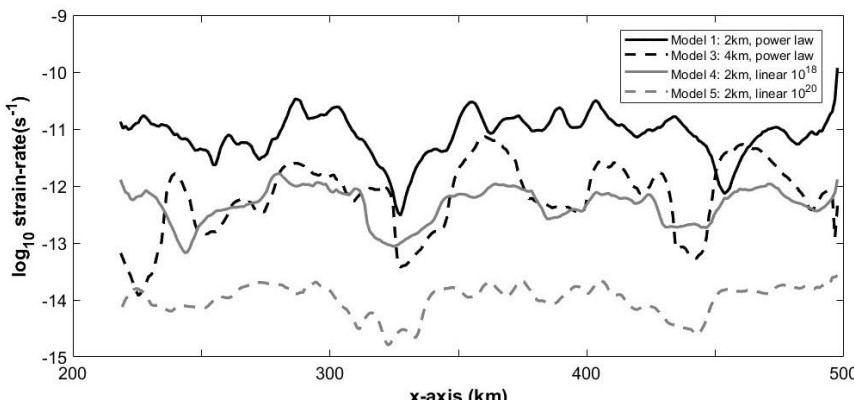

**Figure 8: Vertically-averaged strain-rate over model width for models with variable salt rheology after 15 Myr of shortening.**

## 4.4 Effect of basement fault geometry

Figure 9 shows two models, one with planar basement faults (Model 6) and one with prescribed no basement faults

(Model 7). After 5 Myr of extension, the basement faults maintain a constant dip angle, resulting in the formation of conjugate normal faults rooting in the lower extent of the pre-existing faults (Fig. 9a and S6). Similar to the case with listric faults (Fig. 4c), the second invariant of the strain-rate tensor indicates high concentration of deformation through the development of fractures within the sedimentary layers above the salt horizon, specifically at the back of the footwall of the faults (Fig. 9a and S6). After 15 Myr of compression, multiple overturned forwards and backward

verging folds develop within the sedimentary cover at x ≈ 350 km (Fig. 9b and S6). These folds originate in relation to the activation of thrusts in the underlying basement. Notably, the planar faults play a pivotal role in the formation of backthrust faults. The principal strain rate is concentrated within vulnerable zones such as the salt layer, planar

faults, and the newly formed backthrust faults within the basement (Fig. 9b and S6). Basement faults crosscut the salt décollement to form bypass thrusts.

In the absence of inherited faults (Fig. 9c and S7), no zone of significant strain accumulation is formed during the extension phase. The maximum strain rate accumulates primarily along the fractures caused by extension, while the basement is decoupled from the overlying sedimentary strata. After 15 Myr of shortening, the basement has undergone intense deformation by thrusting (Fig. 9d and S7). Near the backstop wall (x = 350–500 km), it is deformed into pop-up structures caused by conjugate thrusts, while forward verging thrusts dominate towards the foreland. The overlaying sedimentary sequence is mainly shortened by décollement folding/faulting between x = 200–350 km. A large pop-up structure (x ≈ 230 km) without mechanical decoupling between the basement and the sedimentary cover develops in front of the salt pinch-out (Fig. 9d and S7). The strain rates indicate that at the late stage of shortening, a thin-skinned style of deformation is taking place in front of the salt pinch-out and the associated basement thrust.

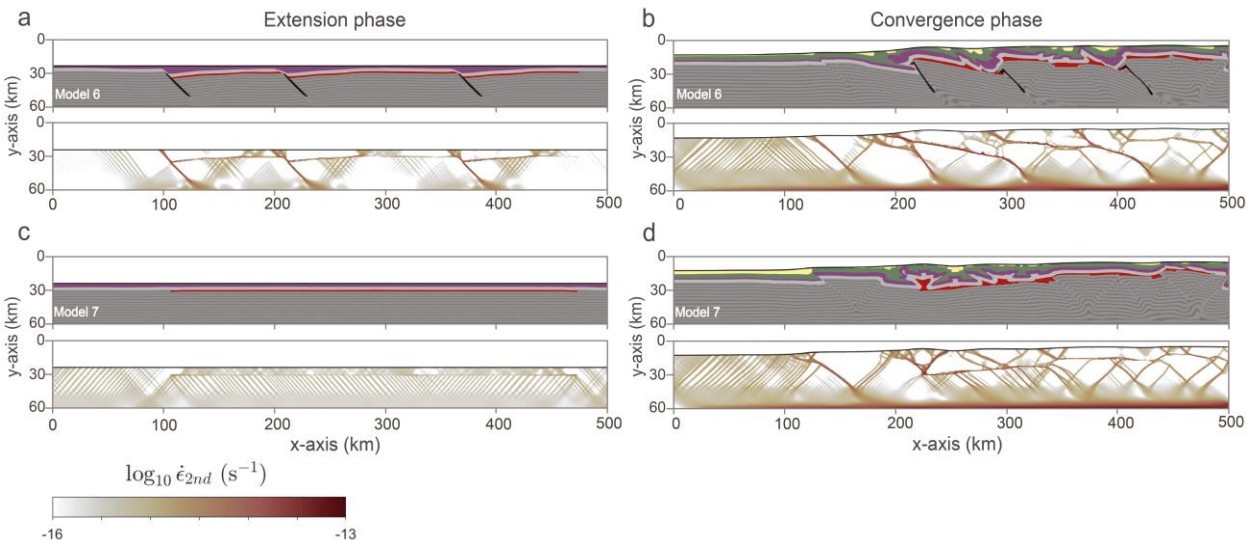

**Figure 9: Composition inferred from Lagrangian markers and the second invariant of the strain-rate tensor, affected by different type of inherited faults after 5 Myr of extension and 15 Myr of convergence. (a-b) In the presence of planar inherited faults. (c-d) in the absence of any inherited faults.**

### 4.5 Effect of basement shortening rate

The effect of basement involvement during convergence is implemented by varying the horizontal velocity of basement rocks exiting the right-side boundary, $v_{x\_b}$ (Fig. 3b). All of the previous models exhibit a rigid backstop over the entire right side that prevented the basement material from leaving the model domain ($v_{x\_b} = 0$), resulting in 100% involvement of the basement in the deformation. In this section, we show a set of models, in which the basement is allowed to exit the model domain with a velocity that is a varying fraction of that on the left boundary (Fig 10). The basement will experience a slower rate of shortening with respect to the cover. As a result, the degree of thin-skinned deformation will increase.

With a basement involvement of 75% ($v_{x\_b} = 0.25 \cdot v_x$), the two faults located on the right side of the model transform into flat-ramp-flat structures (Model 8; Fig. 10a and S8). Above the basement blocks, the sedimentary cover undergoes thrusting and the cover sequence overthrusts the frontal fault zone by ~50 km. If the basement shortening rate is 50%

of the cover shortening rate ($v_{x\_b} = 0.5 \cdot v_x$), the faults situated on the right side of the model remain continuous and

495 undergo significant displacement (Model 9; Fig. 10b and S9). After 15 Myr of convergence, the fault on the right side partly exits the model domain. Additionally, strain rate indicates that most deformation localizes within the salt layer, the pre-existing basement faults, and faults within the sedimentary cover. With a basement involvement of 25% ($v_{x\_b} = 0.75 \cdot v_x$), the fault on the right side almost entirely left the model domain, and the other two faults experience lesser degrees of involvement, roughly displaying 100% of tectonic inversion (Model 10; Fig. 10c and S10). In comparison

to the previous cases, deformation in the sedimentary cover is more distributed laterally, not depending as strongly on the location of basement faults. Furthermore, the salt layer remains connected and forms a nearly-horizontal décollement horizon (Fig. 10c). In the case the basement exits the model domain with the same velocity as the bottom boundary ($v_{x\_b} = v_x$), inherited basement faults are not reactivated, and the sedimentary cover deforms in a thin-skinned tectonic style (Model 11; Fig. 10d and S11). The second invariant of the strain-rate tensor demonstrates that the salt

horizon acts as a décollement layer with basement steps that developed during the rifting phase. The resulting thin-skinned fold-and-thrust belt exhibits a flat surface taper given the weak salt rheology (Fig. 10d).

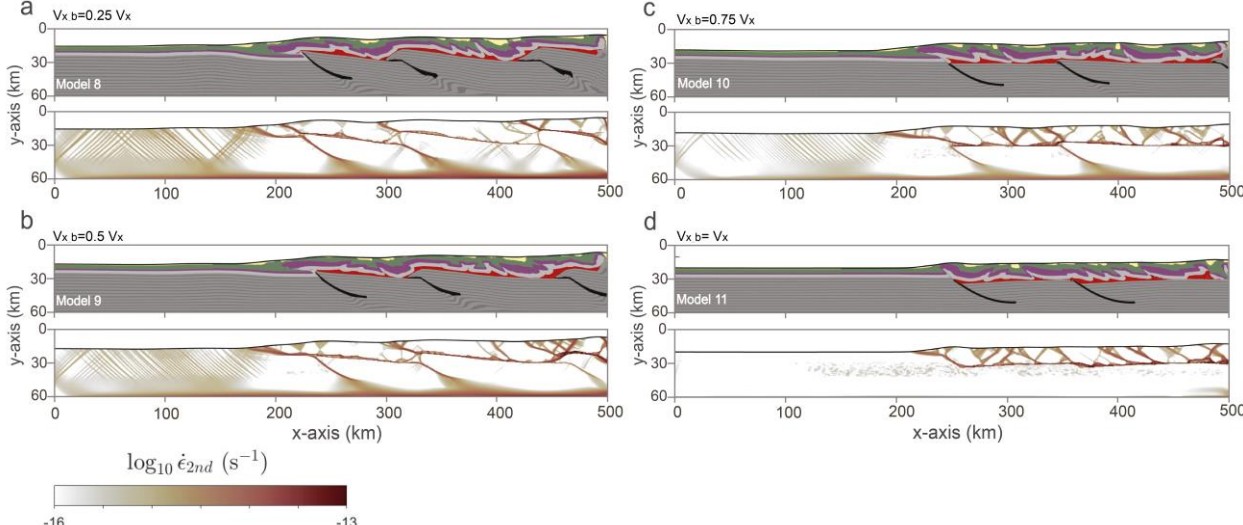

**Figure 10: Composition inferred from Lagrangian markers and the second invariant of the strain-rate tensor, affected by different values of basement velocity after 15 Myr of convergence. (a) $v_{x\_b} = 0.25$, (b) $v_{x\_b} = 0.5$, (c) $v_{x\_b} = 0.75$ and (d) $v_{x\_b} =$**

**1.**

## 5 Discussion

The models presented in the previous sections demonstrate the impact of weak zones, in the form of décollement layers and pre-existing basement faults, and the degree of involvement of the basement on the structural evolution and reactivation of inherited structures during tectonic inversion. In the following, we will provide insight into the effects

of the implemented variables on the dynamic strength of the basement and strain localization within it. Furthermore, numerical results are compared to previous modeling attempts and natural examples from the ZFTB.

**5.1 Strength and localization of deformation within the basement**

The mechanical strength of fold-and-thrust belts and the partitioning of strain during their growth has been a topic of interest for many decades (Chapple, 1978; Davis et al, 1983; Stockmal et al., 2007; Tavani et al., 2015). During continental collision and the emergence of mountain belts, plate convergence rates usually decrease step wise given the increasing strength of the affected plate boundary (Wortel et al., 2009). Weak zones within the deformed rock layers play a crucial role in shaping the structural evolution of fold-and-thrust belts. In the case of the Zagros, numerical experiments demonstrate that both the sub-horizontal salt horizon and inherited weak basement faults significantly influence the partitioning of strain. Regardless of whether the viscous décollement can fully decouple the upper and lower crust mechanically, the basement exerts a strong influence on the overall structural evolution of the Zagros fold-and-thrust belt. This must be considered when constructing structural cross sections.

All experiments with prescribed weak zones within the basement show intense inversion along these inherited structures (e.g. Fig. 4). A particular experiment is Model 7, where no basement faults were prescribed (Fig. 9c,d). The structural style observed in Model 7 is driven by the low rheological strength of the salt layer, which decouples basement shortening from the overlying layers, resulting in thin-skinned deformation of the cover sequence (Fig. 9d). This weak salt layer promotes the formation of fault-propagation folds and conjugate thrusts (pop-ups) rather than a clear structural vergence, as observed by a large pop-up structure at $x \approx 230$ km. The topography of the top of basement is likely caused by a critical wedge geometry defined by the viscous strength of the lowermost modelled crust.

In fold-and-thrust belts with local weak zones in form of décollement layers and reactivated inherited faults, strength may vary across strike. For example, the reference model (Model 1) displays a stress distribution with maximum values in the brittle part of the basement, which is crosscut by hinterland-dipping faults (Fig. 11a). To illustrate across strike variations in strength, three vertical stress profiles are shown representing different segments of the evolving fold-and-thrust belt: i) At x = 150 km, basement faults and a salt décollement are absent, and the strength profile exhibits a simple form with a brittle ($y \approx 15–40$ km) and ductile ($y \approx 40–60$ km) part (Fig. 11b). ii) The second profile at x = 320 km crosscuts both salt décollement ($y \approx 18$ km) and a basement fault ($y \approx 40$ km) that strongly affect the strength profile (Fig. 11c). While the salt horizon shows very low stresses of only a few megapascal, stresses across the basement fault are reduced to ~150 MPa, in contrast to maximum stresses of ~600 MPa around the brittle-ductile transition. iii) Close to the backstop, at x = 460 km, maximal stresses increase to ~800 MPa at $y \approx 40$ km (Fig. 11d). At $y \approx 18$ km, the salt décollement displays an effective decoupling horizon, and an additional weak zone at $y \approx 27$ km is defined by flat ramp of a basement sliver stack. To infer the force acting across the fold-and-thrust belt related to the reference model, the differential stress profiles at each x-coordinate are integrated over the vertical distance. For the reference model after 15 Myr of convergence, values range in between ~7–15 TN/m, where the lowest values coincide with the occurrence of basement faults (Fig. 11c). In general, there is no clear trend observable that would refer to a weakening or strengthening related to deformation, as crustal thickening (strengthening) goes hand in hand with basement thrusting (weakening).

Figure 12 shows the temporal evolution of the average force (e.g. average value of Fig. 11e for the reference model) for a variety of models to identify key parameters that affect the stress state of continental collision zones. All models show similar boundary forces during the extension phase of ~4 TN/m (Fig. 12), except Model 7 without pre-existing

basement faults, which exhibits values of ~4.2 TN/m (Fig. 12b). Similar force values during extension for models with planar and listric inherited basement faults indicates that none of the tested fault geometries would preferably localize. During the convergence phase, boundary forces quickly increase, and most of the models show a general increasing trend from ~6–8 TN/m to ~10–12 TN/m over a time span of 15 Myr (Fig. 12). Minor increase and decrease in force are observed related to the absence (Model 2) and increased thickness of the salt horizon (Model 3), respectively, while the strength (viscosity) of the salt has no detectable effect (Fig. 12a). The absence and geometry of basement faults displays a significant importance during convergence, in contrast to the extension phase, where planar preexisting faults increase the necessary boundary force by ~1–2 TN/m, and no faults lead to another increase of ~1–2 TN/m during the first 10 Myr of convergence (Fig. 12b). More important is the effect of basement involvement in crustal shortening, with decreasing involvement resulting in decreasing boundary force (Fig. 12c). This illustrates the importance of an effective décollement level that is able to decouple an upper crust with distributed deformation from a basement that is underthrust towards the suture, where it deforms more intensely (Tavani et al., 2015; Pfiffner et al., 2002). A similar pattern of boundary force evolution was also observed from analogue models, however, with more acute variations for specific internal deformation of the compressed sand pile (McBeck et al., 2018).

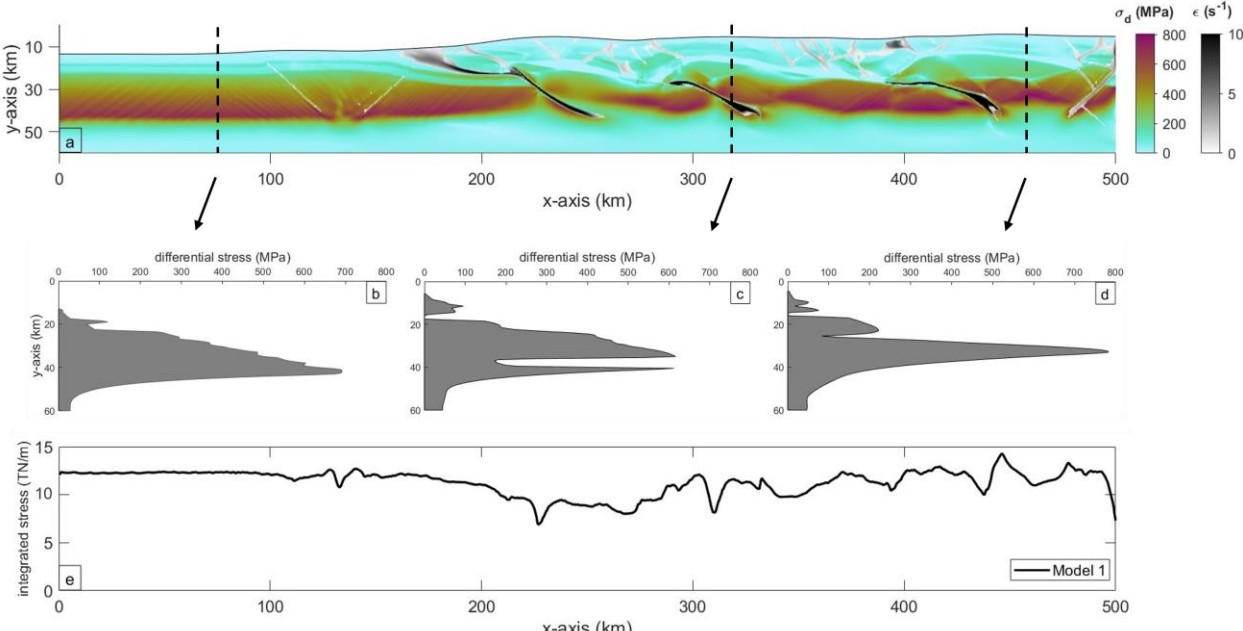

**Figure 11: (a) strain tensor superimposed on differential stress tensor. (b-d) Differential stress profiles of front of the deformation front, with decollement and basement fault, and without basement fault involved respectively. (e) The integrated stress profile for reference model.**

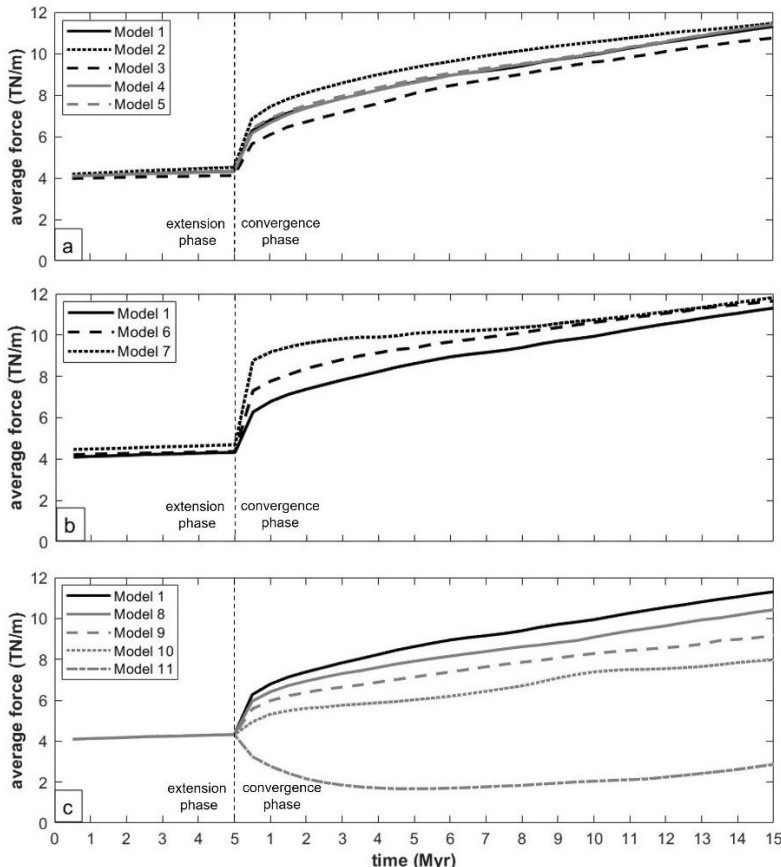

**Figure 12: Horizontally-averaged boundary force over time. (a) Models with variable salt thickness and salt rheology. (b) Models with variable fault geometry. (c) Models with variable basement shortening rate. Model characteristics are listed in Table 2.**

## 5.2. Comparison with numerical modelling studies

The mechanics of fold-and-thrust belts has been investigated by means of numerical methods since the 1980's (e.g. Borja and Dreiss, 1989). Rapidly evolving computing capabilities led to a vast increase in geodynamic numerical modelling studies that provide an invaluable contribution towards our understanding of Geotectonic systems (e.g. van Zelst et al., 2021; Gerya, 2022; Ismail-Zadeh and Tackley, 2010). Previous numerical models of fold-and-thrust belts have typically focused on thin-skinned tectonic systems, investigating parameters such as wedge and décollement strength, rheology, surface processes, and mechanical stratigraphy (e.g. Stockmal et al., 2007; Buiter et al., 2016; Simpson, 2011; Burbidge and Braun, 2002; Ruh et al., 2012). However, the inclusion of rifting prior to collision adds a new dimension that has often been overlooked in these studies. This approach allows for a more comprehensive understanding of tectonic evolution by examining how pre-collisional structural configurations influence later deformation patterns. Including rifting before convergence is crucial as it sets the initial conditions that significantly impact structural evolution during collision (Buiter and Pfiffner 2003; Ruh et al., 2018; Granado and Ruh, 2019). This approach offers several advantages: It provides insight into structural inheritance, as rifting creates pre-existing weaknesses and fault systems that play a crucial role during subsequent compressional phases. Our model demonstrates how these inherited structures influence strain localization and deformation styles, providing insights

into the evolution of complex geological features such as the Fars Arc. In the following, we discuss several numerical studies concerning tectonic systems that are similar to the one presented in this study to set our results into perspective and to identify generally applicable observations. We, furthermore, comment on the importance of implementing an extensional phase into the model setup.

Buiter and Pfiffner (2003) applied a two-dimensional, visco-plastic numerical model in order to evaluate the dynamics of tectonic inversion of a series of half-graben basins upon compression. They reported syn- and post-rift sediment uplift, accompanied by basement block rotation, and the emergence of newly formed shear zones in the post-rift sequence, originating from basin-bounding faults. Weak sediments at the basin base contribute to the generation of basement shortcut faults. Furthermore, their study illustrates the predominant development of back thrusts as
conjugates to listric basin-bounding faults. The structural evolution observed in our models aligns with their research findings, demonstrating a similar pattern of strain localization within syn-rift and post-rift layers, mainly initiated by deformation along pre-existing underlying faults. Additionally, to their findings, our results indicate that the formation of back thrusts is mainly associated with the inversion of planar faults (Fig. 9; Model 6).

Nilforoushan et al. (2013) presented two-dimensional thermo-mechanical experiments of thick-skinned fold-and-
thrust belts with salt décollements. They demonstrated that the geothermal gradient and the mineralogy of the basement (ductile flow law) strongly affect the geometry and reactivation of inherited basement faults. Furthermore, they underlined the importance of a weak salt horizon at the base of the sedimentary succession for the mechanical decoupling and formation of a thin-skinned upper-crustal fold-and-thrust belt, resulting in a variability in the rate of shortening between the cover and the basement. Their findings corroborate the hypothesis of Molinaro et al. (2005)
that in the ZSFB, greater shortening occurs within the cover compared to the basement. From our results examining the effects of the shortening rate of the basement on structural development, we observed the emergence of long-wavelength folds arising directly from the movements of faults within the basement (Figs. 11a-d). These findings align with natural occurrences and are corroborated by geological cross-sectional data, reinforcing the principal characteristics of the Fars arc that are typified by folds of extensive wavelengths. This agrees with the suggestions by
Mouthereau et al. (2006), who proposed that early activation of basement deformation dictates the formation of the large-wavelength folds. Furthermore, Mouthereau et al. (2006) determined that the viscosity of Hormuz salt decreases significantly with increasing temperature, exhibiting Newtonian fluid behaviour under low stress and strain-rate conditions. In our results, the average viscosity of non-Newtonian salt is affected by both temperature and strain rate, where weak zones with the lowest strain rates and temperatures exhibit the largest viscosity values. The resulting
values are larger than what was proposed for the Fars arc resulting from Critical Wedge Modelling (Mouthereau et al., 2006).

Ghazian and Buiter (2014) investigated the impact of salt for the southeast ZFTB by 2D thermo-mechanical models. Their findings revealed that the presence of a thick basal Hormuz salt effectively decouples overlying sediments from the basement and promotes the localization of deformation in the sediments. Our results align with these findings,
demonstrating a significant influence of the salt layer on controlling topographic height and folding patterns (Fig. 4 and 6). The deformation style in both the basement and the cover sequence decouples along the weak salt horizon. As

the salt thickness increases, this layer functions more effectively as a décollement, impeding deformation in the sedimentary cover sequence (Fig. 6).

Bauville and Schmalholz (2015; 2017) and Kiss et al. (2020) conducted 2D numerical experiments to investigate the transition from basement-involved thin-skinned to thick-skinned tectonics and the effects of tectonic inheritance on the development of nappe systems. The key parameters controlling this transition were found to be the viscosity ratios within the basement and between the basement and the sedimentary cover above it. Specifically, a higher ratio within the basement favors thick-skinned deformation, while a higher ratio between the basement and cover leads to thin-skinned deformation. As indicated by our study, 100% participation of the basement in deformation leads to thick-skinned deformation (Fig. 4), while non-participation of the basement to thin-skinned deformation (Fig. 10; Model 11). Their findings show the tectonic and structural inheritance plays a significant role in controlling the tectonic evolution and resulting structures in the fold-and-thrust belts. The geometry and magnitude of mechanical heterogeneities like varying basement-cover interface characterized by half-grabens and horsts, as well as the vertical alternation of sedimentary layers with different mechanical strengths, influence nappe formation. Their observations align with our findings that the presence of pre-rift salt and inherited faults, which serve as structural inheritances, influence the shape and geometry of the basin (Fig. 4,6,7). Moreover, their simulations and ours revealed that both linear and power-law viscous rheologies demonstrated similar features to more complex simulations, indicating the robustness of these findings (Fig. 7; Model 4 and 5).

Eslamrezaei et al. (2023) used numerical discrete element models to investigate the impact of mechanical stratigraphy, décollement layers, and the number and thickness of cover sequences on the structural evolution and strain partitioning of thin-skinned fold-and-thrust belts. Their modeling outcomes revealed that weak décollements played a crucial role in decoupled deformation within the fold-and-thrust belt under continuous shortening. The study underscored that shortening was primarily accommodated by thrust-related folds, resulting in notable variations in structural styles. In alignment with our findings (Fig. 4, 6 and 10), the weak décollement layer, due to its characteristics, exhibited an inability to retain stress and undergo deformation, as evidenced by the increasing strain rate along the basal detachment (Fig. 4). Both their study's conclusions and our results emphasized that the structural style and decoupling in thin-skinned fold-and-thrust belts and formation of structures such as box folds and harpoon structures, are influenced by factors such as rheology, the number, and thickness of décollements (Fig. 6 and 7).

An essential characteristic of the numerical experiments presented here is the implementation of a rifting phase prevailing crustal shortening. Earlier numerical studies on tectonic inversion demonstrated that strain-related weakening of the crustal basement during an extensional tectonic phase has an important effect on the localization of deformation during consequent convergence (Ruh and Vergés, 2018). Furthermore, the mechanical strength of sedimentary deposits filling rift-related basins has been shown to influence the structural evolution during tectonic inversion, where weak syn-rift deposits favour the development of hanging wall by-pass structures (Granado and Ruh, 2019). In contrast, various studies mimic the extensional phase by prescribing inherited structures and geometric configurations comparable to rifted margins instead of conducting an extensional phase when investigating the tectonic inversion of fold-and-thrust belts (e.g. Buiter and Pfiffner, 2003; Nilforoushan et al., 2013; Bauville and Schmalholz, 2015; Kiss et al. 2020). However, the implementation of an extension phase allows for a dynamic

formation of basement steps and graben geometry based on the thermo-mechanical model characteristics such as the visco-plastic/brittle transition and the evolution of the salt decoupling along the basement top (Fig. 4a-c and 5a,b). Model 7 demonstrates that a certain way of strain localization is needed to form narrow shear bands similar to basement faults during extension (Fig. 9c). While other studies impose specific boundary conditions (Ruh and Vergés, 2018) or thermal variations (Ruh and Vergés, 2018), we introduced predefined geometries to localize extensional normal faults, controlling the position of basement deformation (Fig. 3b).

## 5.3 Comparison with the structural evolution of the Fars Arc

The tectonic style of the Fars Arc is characterized by a combination of thin-skinned salt-related folding and thick-skinned basement thrusting (i.e., Mouthereau et al., 2007a). The structure of the sedimentary cover is shaped by a series of detachment faults and fault-propagation folds, involving several salt diapirs rooted in the Hormuz salt (e.g. Jahani et al., 2007; Motamedi and Gharabeigli, 2019; Vergés et al., 2024). In addition, at least three high-angle reverse basement faults (High Zagros, Surmeh, and Mountain Front faults) are constrained based on their seismicity, structural relief, magnetic anomalies, and abrupt topographic changes (e.g. Jackson and Fitch, 1981; Berberian, 1995; Talebian and Jackson, 2004; Mouthereau et al., 2007b; Teknik and Ghods, 2017; Karasözen et al., 2019). The dynamic importance of these faults is supported by our results, as numerical models without basement faults (Model 7 in Fig. 9) display a marked difference in structural style with geological observations in the Fars Arc. In the absence of basement faults in the models, deformation is concentrated only in the outer part of the arc, in the form of an imbricate thrust system and a large mushroom anticline near the salt pinch-out. Furthermore, the importance of the Hormuz salt for the distinct structural style of the Fars Arc is corroborated by the numerical model without a basal salt layer (Model 2 in Figs. 6a and 6b). In that model, four basement-involved anticlines form across the entire belt, each with wavelengths of about 100 km, while in contrast, in the Fars Arc, the anticlines can be grouped into short-wavelength detachment-fold anticlines and long-wavelength basement-involved anticlines (e.g Mouthereau et al., 2006). In the model without salt, all structures verge towards the front without any back-thrusting. This type of fold and faults geometry is not representative of the observed structures in the Fars Arc (e.g. Motamedi et al., 2012; Najafi et al., 2014; Jahani et al., 2017). Observational evidence thus highlights the essential role that the basement and the salt layer play in determining the deformation style of the belt.

The results of the experiment with a 4-km-thick basal salt layer (Model 3; Fig. 6c-d) demonstrate that a thicker salt layer interacts with the basement faults to form symmetrical anticlines containing salt cores and discordant diapirs. The topography of the Fars Arc cannot be solely attributed to the presence of a salt layer, the basement's role in structural development is also significant. As our results indicate, processes involving the basement should be considered as an essential factor in reconstructing and assessing the geological development of the region. The models demonstrate that the basement faults ramp up through the upper part of the basement and eventually flatten along the Hormuz salt level at a depth of around 10 km (Model 1 in Fig. 4f). Similar structures of basement faults detaching along evaporitic levels in the middle of the sedimentary cover, or reaching the surface to cut the forelimbs of their hanging wall anticlines, are documented in the Fars Arc (Vergés et al., 2024), such as the Kangan, Asaluyeh and

Tabnak anticlines, located to the south of the cross-section in the foreland vicinity and within the Mountain Front Fault (Berberian, 1995; Mouthereau et al., 2007b; Figs. 13b,c). These anticlines are characterized by a pronounced concentric surface geometry (Fig. 13a).

A well-studied example of syntectonic growth strata is the Dowlatabad syncline, located approximately 40 km northeast of the Persian Gulf coastline (Figs.13a, b) and limited by the Sefid anticline to the north and the NW segment

of the Pazan anticline to the south (Najafi et al., 2021). The Dowlatabad syncline is infilled by more than 2 km of syn-folding sediments. Numerical results from Model 3, which has a thick salt layer, compare  well with the formation of the Dowlatabad syncline (Figs. 13a,b): i) The forelimbs of Pazan and Sefid anticlines were faulted by foreland-directed thrusts rising from the basal décollement, ii) surface processes resulted in a smooth topography, and iii) the surface has an asymmetric shape, while at depth, the structure has the form of a box fold.

In the Zagros belt, there is a noted relationship between faulting patterns and the distribution of salt diapirs (Talbot and Alavi,1996; Hessami et al., 2001; Bahroudi and Koyi, 2003; Sherkati and Letouzey, 2004; Sepehr and Cosgrove, 2005; Jahani et al., 2017). Our numerical results indicate that salt diapirism may be triggered at two distinct stages in the case of a thick (4 km) salt décollement. During the rifting phase, small-scale diapirs start to grow above the tip of normal faults, partially piercing the sedimentary cover (Model 3; Fig. 6c). This procedure can be considered as a short-

lasting and locally-developed reactive diapirism stage (e.g. Jackson and Vendeville, 1994; Jackson and Hudec, 2017). More significantly, the growth of two salt diapirs are observed during the compression phase in the inner zone of the fold-and-thrust belt (Model 3 in Fig. 6d). In contrast to décollement folds, these salt bodies have pierced the overburden strata and show discordant contact with overlying sediments. The process involves the buoyant rise of underground salt when subjected to pressure from convergent tectonic forces. These early-growing diapirs, squeezed

by further shortening, lead to the development of secondary salt welds. This type of shortened diapirs has recently been documented in the High Zagros zone in a field-based study by Taghikhani et al. (2024) and have been already known in the SE Fars Arc (e.g. Callot et al., 2007; Jahani et al., 2007) and the Persian Gulf (e.g. Hassanpour et al., 2020, 2021; Snidero et al., 2020) through seismic interpretations. A well-exposed example of such a structure is the Hajiabad diapir, located in the inner part of the Fars Arc, where an 8 km-long squeezed Hormuz salt diapir trends NW

and dips towards NE (Fig. 13d). The salt-bearing evaporites have been shortened between the two flanks of the precursor salt wall, forming a secondary salt weld. Another well-exposed example in the inner Fars is the Kharman diapir (Fig. 13e), which is in good correlation with observations from Model 3. Fernandez and Kaus (2014) showed with numerical modeling that pre-existing salt diapirs can significantly influence the pattern and growth of three-dimensional folds and fold patterns, accelerating fold formation and localizing deformation, highlighting the important

role of diapirism in structural evolution during tectonic processes.

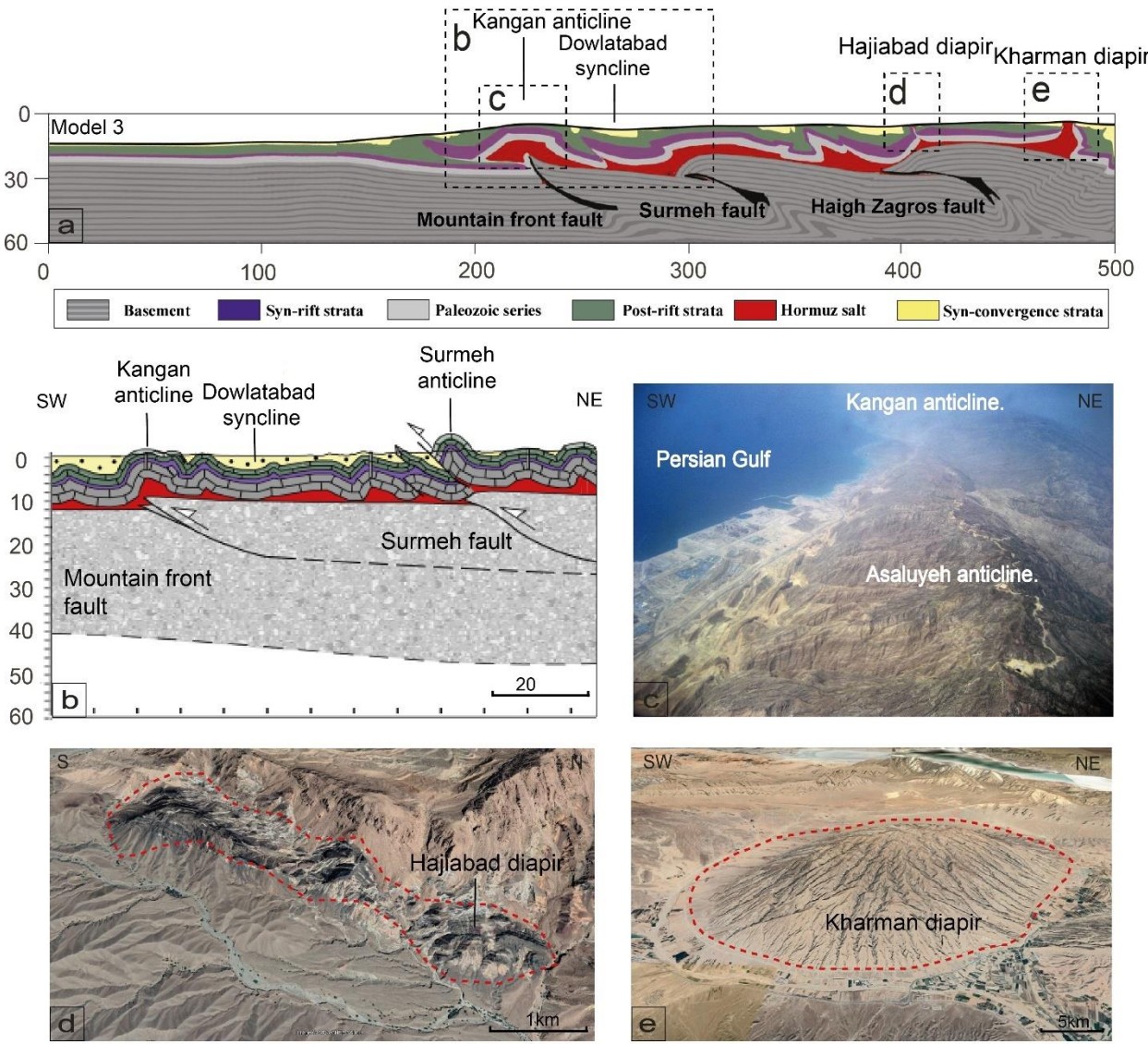

**Figure 13: (a) The result of numerical modeling with thick salt and basement involved (Model 3). (b) Geological cross section of the Dowlatabad syncline and Kangan anticline (modified from Najafi et al. 2021). (c) Aerial image of Asalouye and Kangan anticlines. (d) Satellite image of the Hajiabad Diapir (© Google Earth). (e) Satellite image of the Kharman Diapir (© Google Earth).**

## 5.4 Implication for seismic activity in the Fars Arc

Figure 14a displays a seismicity map of the Zagros in the region of our profile. It shows events with magnitudes greater than $M_L = 4$, between 2000 and 2023 from the ISC global database (http://www.ics.ac.uk). We have excluded those events of the database with pre-fixed depths (usually set at 10 or 15 km) from the plot. Furthermore, a catalogue of small-magnitude earthquakes from Tatar et al. (2004) recorded by a local temporary network over a period of seven weeks is plotted. A local network with close station spacing allows for more precise location of earthquakes, and Tatar et al. (2004) state that these events have horizontal and vertical uncertainties below 2 km. The earthquakes are also plotted onto the geological cross-section of figure 1c and the compositional image of the final stage (15 Myr) of Model 1 (Fig. 14b, c). Several studies have asserted that the depths and focal mechanism solutions indicate that high-angle

reverse faults in the basement are responsible for major earthquakes in the SE Zagros (e.g. Jackson and Fitch, 1981; Berberian, 1995; Talebian and Jackson, 2004). Others have argued that most of the strongest earthquakes in the ZSFB probably lie in the lower sedimentary cover, but their aftershock activity mostly occurs deeper in the basement (e.g. Nissen et al., 2011; 2014). A more recent work by Karasözen et al. (2019) used calibrated earthquake relocation to show that the Zagros earthquakes nucleate in both the cover and the basement. According to our numerical results,

the activation of basement structures localizes topographic features and affect thrust faults within the overlying sedimentary cover (Fig. 4f, 9b, 10). Notable examples of active basement faults in the Fars Arc include the Surmeh Fault and the Mountain Front Fault, which have distinct structural and topographic characteristics (Jackson,1980; Jackson and McKenzie, 1984; Berberian, 1995; Mouthereau et al. 2007b). Critical wedge modeling confirms that basement-involved shortening is the primary factor influencing deformation and topography in the Fars Arc (Fig. 4f

and 9b; Mouthereau et al., 2006). Balanced cross-sections provide additional evidence of the basement's involvement in shortening through major thrusts rooted in a deep décollement level within the lower crust (Molinaro et al. 2005; Mouthereau et al, 2007; Najafi et al; 2014; Najafi et al; 2021).

       The majority of earthquakes are concentrated within the sedimentary cover, aligning with the décollement faults (Fig. 14b, c). This observation supports the theory of the sedimentary cover decoupling from the basement, influenced by

the presence of the basal salt layer. The depth trend of earthquakes in the basement exhibits a notable correlation with the High Zagros Fault and the Surmeh Fault, which confirms the existence of pre-existing weak zones within the basement at these locations. Furthermore, the depth distribution of the earthquakes supports the idea of a listric geometry for the basement faults. By examining the deformation within the basement and the earthquake depths, we estimate the transition zone from brittle to ductile behaviour at around 30 km. While this depth is influenced by

multiple factors, including material parameters, temperature, and strain rate (stress), our approximation is based on typical conditions relevant to the region. This suggests that diabase rheology, combined with the applied geotherm, is appropriate for modeling the basement under these specific conditions. Mouthereau et al. (2006) on the other hand concluded that diabase might be too weak to reproduce the observed topography in the Fars arc. However, they apply a crustal thickness of 45 km, which implies increased temperature conditions and thus a weaker diabase detachment.

Our modeling results indicate that reverse faults cutting both flanks of décollement anticlines are likely responsible for the shallower earthquakes (Fig. 14c). This phenomenon was studied in the Shanul anticline, southeast of our study region, where earthquakes with reverse mechanisms occur on both flanks of the anticline at depths mostly ≤4 km (Jamalreyhani et al., 2021). However, basement faults are the source of deeper and larger earthquakes across the Zagros fold belt, generally concentrated in NE-dipping linear trends (e.g. Nissen et al., 2019; Fig. 14).

Lacombe and Mouthereau (2002) discussed the relative chronology between mid-crustal décollement thrusting and shallow décollement folding of sedimentary cover in various orogenic belts, such as Taiwan, the Pyrenees, and the Alps. This relationship has been further investigated through analogue and numerical modeling in the Jura, Appalachians, Kopet Dagh, and Zagros fold-and-thrust belts (e.g. Pohn, 2000; Orjuela et al., 2021; Ruh and Vergés, 2018; as presented in this study). A comprehensive analysis of these studies, in alignment with our results, suggests

that there is no definitive sequence for thin- and thick-skinned tectonics in all orogenic belts. This sequence appears to be controlled, at least in part, by the rheology and temperature of basement rocks, the thickness of evaporitic layers

in the sedimentary cover, the dip and flatness of inherited basement faults, as well as their orientation in relation to the tectonic transport direction.

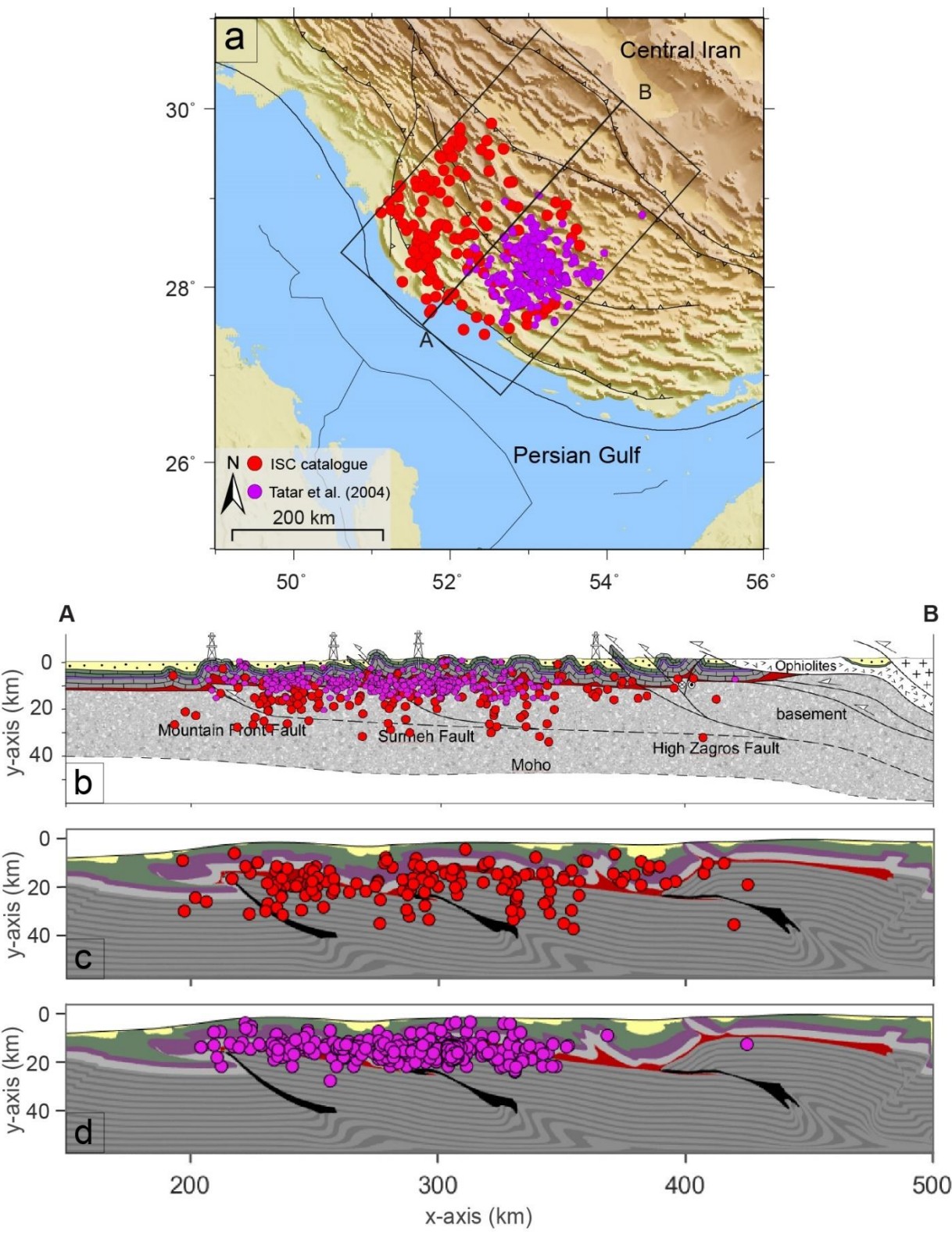

**Figure 14: (a) Seismicity of Zagros fold-and-thrust belt reported by ISC during the years 2000 to 2023 with magnitudes $M_L > 4$ and local temporary network data by Tatar et al (2004) with magnitudes ML<4, superimposed on a shaded relief map derived from Global Topography. (b, c, d) Projection of earthquakes along profile A–B after removing fixed depths on geological cross section and the reference model (Model 1) after 15 Myr of convergence.**

## 6 Conclusions

Our series of 2D finite-difference thermo-mechanical numerical experiments provide insight into how the rheology of basal salt and the mechanical properties of inherited basement faults control the structural evolution and final style of fold-and-thrust belts. The models were designed based on the tectonic history (continental rifting and later collision) and mechanical stratigraphy of the Fars Arc in the Zagros orogenic belt. The comparison of numerical results with the actual structure of the Fars Arc has led to the following conclusions.

1) Pre-existing weak zones, such as basement faults and décollement anticlines, serve as primary sites for deformation accumulation, particularly during extensional tectonic phases.

2) The basement faults form large-wavelength foreland-verging fault-propagation anticlines in the overlying sedimentary cover, while the thick salt layer promotes the growth of second-order smaller-wavelength décollement anticlines accompanied by both fore-and back-limb thrust faults.

3) Reactivated faults play a crucial role in stress transfer, leading to the formation of new faults and seismic activity, at their tips propagating into the sedimentary cover. Listric faults are more effective at accommodating strain rates.

4) The presence of a basal salt layer influences fault displacement, contributing to the development of back thrusts within the sedimentary cover.

5) The distribution of earthquakes is significantly influenced by the presence of weak zones, with shallow earthquakes predominantly occurring along décollement levels and larger, deeper earthquakes associated with basement faults. These findings highlight the importance of considering geological and structural complexities when assessing seismicity and tectonic events in tectonically active regions like the Fars Arc.

6) The degree of basement involvement directly influences the model's resistance, with greater involvement facilitating deformation process over geologic time. In addition, variations in resistance to deformation based on salt rheology and fault geometry were observed, with listric faults minimizing resistance and therefore facilitating deformation.

7) Intense basement deformation observed in the numerical experiments indicates the importance of the lower crust for the construction of regional cross sections of the Zagros fold-and-thrust belt.

## Author contribution

FG: modeling setup, experiments run, investigation, writing – original manuscript and handling, visualization, figure drafting. JR: hypothesis of work, numerical modeling code write up, interpretation of results, validation, supervision, manuscript writing, and reviewing and editing. MN: hypothesis of work, interpretation of results, supervision,

fieldwork and cross-section construction, reviewing and editing. FS: hypothesis of work, modeling setup, supervision,
and reviewing and editing. All authors contributed and approved the submitted article.

**Competing interests**

The authors declare that they have no conflict of interest.

**Acknowledgments**

We would like to express our sincere gratitude to the reviewers, Dr. Frédéric Mouthereau and Dr. Lorenzo Giuseppe
Candioti, for their time and effort in reviewing our paper. Their insightful comments and suggestions have
significantly improved the quality of the manuscript. Additionally, we extend our appreciation to Dr. Susanne Buiter
for her comments and editorial guidance throughout the review process. This manuscript is based on the Ph.D.
dissertation of the corresponding author, Fatemeh Gomar, at the Institute for Advanced Studies in Basic Sciences
(IASBS), Zanjan, Iran. It was partly developed during her 6-month scientific stay in the Structural Geology and
Tectonics group at ETH Zurich, Switzerland.

**Data availability**

The GIF movies of all experiments presented in this study are available in Figshare with the identifier:
https://figshare.com/s/cb0c7f5a9a6fda0e38ef

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
