# Peer review of "Importance of basement faulting and salt decoupling for the structural evolution of the Fars Arc, Zagros fold-and-thrust belt: A numerical modeling approach"

_EGUsphere, 2024_

## Author Response (AR1)

**##General comments**

*This study aims to investigate using a numerical modelling approach the coupling between basement-involved shortening and salt-based cover deformation and its impact on the tectonics*

*style of the Zagros fold-thrust belt. One of the novelty of this research is to implement the evolution of the basement/cover shortening in the Zagros from rifting during the NeoTethyan rifting to mountain building, and the exploration of a useful range of parameters including variations of forces and strain rates across the fold-and-thrust belt. The manuscript is well written and organised. However, I don't think that they are yet sufficiently discussing the*

*implications of their results in regard to the few previous studies, for instance, how the topographic slopes they predict conform with observations as the models do not account for flexure?*

> *We would like to thank Reviewer #1 (Frédéric Mouthereau) for careful reading and constructive review of the manuscript.*

> *Thanks for the comment. We outline the fact that flexure has been ignored in the model setup and its potential effect on surface tapers: "The finite Eulerian domain impedes elastic bending of the lower boundary, affecting resulting surface taper angles compared to the Zagros fold-and-thrust belt (Mouthereau et al., 2006; McQuarrie, 2004)."*

*Moreover, although the references chosen are generally sounding, they should refer to the earlier studies on the Fars arc in the introductory part of the study not only in the discussion, so the reader can better understand what has been done on the subject.*

> *The introduction section has been revised, and we added a paragraph that mentions previous studies in the field of numerical modeling and geology related to the Fars Arc*

> *or other fold-thrust belts:*

> *"Mouthereau et al. (2006) reported that basement deformation and thickening is crucial in the Zagros Folded Belt to reproduce the topographic growth observed. They furthermore highlight that the early stage of compression in the Zagros foredeep was controlled by the reactivation of pre-existing faults, suggesting that inherited structural*

> *features significantly affect current deformation patterns."*

> *"Various 2D numerical modeling studies have investigated the evolution of fold-and-thrust belts and salt-bearing basins. Nilforoushan et al. (2013) demonstrated the influence of geothermal gradients and basement mineralogy on fault geometry and basement reactivation in the Fars arc, emphasizing the role of weak salt horizons in*

> *mechanical decoupling. Heydarzadeh et al. (2020) analyzed factors such as sedimentation rates, erosion, and salt layer properties in Dehdasht basin, highlighting the importance of balanced surface processes and deformation rates. Humair et al. (2020) conducted simulations to study the interaction of folding and thrusting during Swiss Jura and Canadian Foothills fold-and-thrust belt evolution, focusing on the effects*

> *of layer-parallel shortening and initial geometrical perturbations. Their work showed that the magnitude of these perturbations influences whether folding or thrusting predominates, affecting the structural evolution and asymmetry of anticlines. Spitz et al.*

*(2020) conducted 3D thermo-mechanical numerical simulations to investigate the influence of laterally variable inherited structures on fold-and-thrust belt evolution and nappe formation on Helvetic nappe system. The study demonstrated the fundamental importance of tectonic inheritance on fold-and-thrust belt evolution, with strain localization, folding, and nappe transport controlled by initial geometrical and mechanical heterogeneities. Almost all studies have focused on examining the collisional phase and deformation resulting from compression in the fold belts and the Fars Arc, while the earlier extensional history and its effect on later deformation have received less attention (e.g., Granado and Ruh, 2019)."*

*Finally, I would have like to see how this work can help drawing better balanced cross-section sections. For instance, to account for decoupling in the basement geologists used to draw either a detachment in the middle crust or in the lower crust. According to my reading of the model results, the upper-middle crust appears progressively decoupled from the lower crust so that we see distributed shearing in the lower crust, meaning a significant part of the lower crust is indeed involved in the deformation. One consequence is that geologist should not draw a localised detachment in middle crust but rather a distributed zone of deformation or a detachment deeper in the lower crust to account for the lower crustal material involved. This point is only technical because it has implications in terms of crustal budget during convergence. Find more detailed comments below.*

*That is correct; a localized detachment in the middle crust is not favoured. However, the deformation in the lower crust is to a certain degree pre-defined by the velocity by which the basement is allowed to leave the model domain on the right, which is one of the parameters investigated in this paper (Fig. 10). There, the strain rate plots clearly indicate the change from an interrupted detachment (a) to a sub-horizontal uninterrupted detachment (d). We added a sentence to the discussion and the conclusions, pointing out the importance for cross-section interpretation:*

*"Weak zones in the deformed rock pile define the structural evolution of evolving fold-and-thrust belts. According to the presented numerical experiments, both the sub-horizontal salt horizon and inherited weak basement faults strongly affect strain partitioning in the Zagros, eventually. Independent of whether the viscous décollement is able to decouple the upper from the lower crust mechanically, the basement has a strong effect on the overall structural evolution of the Zagros fold-and-thrust belt, which has to be taken into consideration when attempting the construction of structural cross sections."*

*"Intense basement deformation apparent in the presented numerical experiments indicates the importance of the lower crust for the construction of regional cross sections of the Zagros fold-and-thrust belt."*

**##Specific comments**

*Lines#63-65: Alternatively looking carefully at the Mountain Front Fault, which is the topographic front of the Fars arc (Asaluyeh anticline), there are evidence that it is related to basement thrusting. First, the topographic offset between for the forelimb and backlimb is best explained by a basement fault. Then, the Fars arc is devoid of salt diapir (they are exposed only across the basement faults) compared to adjacent areas. It is also characterized by deep*

*earthquake that is best explained by heterogeneities in the basement (Mouthereau et al., 2006; Mouthereau et al. (2007). So the arc shape is not simply the result of a fast propagation caused by salt. We showed that the regional topography was the result of basement thrusting because the salt was to weak to reproduce the topography of the Fars arc.*

*According to your suggestions, the text has been revised and the references have been updated:*

*"The presence of a thick salt layer at the base of the sedimentary cover in the Fars Arc is responsible for short-wavelength folds (Mouthereau et al., 2006)."*

*"In the Fars Arc, the activity of inherited faults has affected the progression of deformation towards the foreland, with the Mountain Front Fault being related to basement thrusting (Bahroudi and Koyi, 2003; Mouthereau et al., 2006; Mouthereau et al., 2007a; Yamato et al., 2011; Ruh et al., 2014; Najafi et al., 2021)."*

*Line#73: Please cite previous works that provided more quantitative estimates in support of basement-involved faulting (Mouthereau et al., 2006; 2007). We also suggested that both thin and thick-skiined deformation occurred synchronously.*

*The references have been updated.*

*Line#95: If you are talking about orogen and collision I think Mouthereau et al. 2012 is a better*

*The references have been updated.*

*Line#114: this is more or less the same team. You should also cited older works conducted in the Fars arc I think like Khadivi et al., 2009 or the synthesis in Mouthereau et al., 2012.*

*The references have been updated.*

*Line#163: cite Mouthereau et al. 2007; 2012 who provided one of the first cross-section and kinematic analysis for the Fars arc.*

*The references have been updated.*

*Line#165: same here cite Mouthereau et al., 2012.*

*The references have been updated.*

*Lines#236-238: 600°C at 30 km correspond to 20°C/km. Isn'it too low ? Perhaps the fact that the LAB below the Zagros is supposedly thick (see Tunini et al., 2014) might help justify this. Additionally does this fit with the thermal age (Neo-Tethyan) you expect for the margin?*

*Thank you for your valuable feedback, as you mentioned, the lithospheric thickness beneath the Zagros is significantly thicker than in Central Iran. The study by Tunini et*

*al. (2014) concludes that the lithospheric thickness beneath the Zagros fold-and-thrust belt is thick, reaching around 200 km, and decreases towards Central Iran. This is further supported by other studies by Jiménez-Munt et al. (2012) and Priestley et al. (2012), which estimated the LAB depth to be around 180-220 km. The thick lithosphere beneath the Zagros acts as an insulator, hindering the transfer of heat from the mantle to the crust. As a result, the temperature gradient in this region is relatively low. Furthermore, Aldega et al. (2018) inferred a constant geothermal gradient of 20 °/km in the Zagros based on paleothermal data. We added the following statement at the respective place in the text:*

*"The initial temperature field is characterized by a linear temperature increase with depth, starting from 0 °C at the interface between rock and sticky-air and reaching 600 °C at the bottom, in agreement with a constant geothermal gradient of 20 °/km inferred from paleothermal data (Aldega et al., 2018) and a relatively thick lithosphere beneath the Zagros (Jiménez-Munt et al. 2012; Priestley et al. 2012; Tunin et al., 2014)."*

*Line#263: this is little more than the 16-19% we estimated in the centre of the Farc arc. We also estimated a shortening rate of 6.5-8 km/Myr; are these values consistent altogether ?*

*Thanks for the comment. As showed in previous studies, on average, the overall shortening in the Fars Arc can be considered equivalent to roughly 20% (McQuarrie, 2004; Mouthereau et al 2007; Motamedi et al., 2012; Pirouz et al., 2017; Najafi et al., 2021). 6-8 km/Myr seems to be rather fast taking into consideration recent GPS velocities and total shortening. However, rates of the mountain front propagation might reach such high values. We added references to support our choice of total shortening:*

*"The compressional phase ran for 15 Myr, accommodating a total shortening of 25% (i.e., 125 km). This amount offsets the 25 km extension during the extensional phase and the 20% shortening during the collisional phase, comparable to estimates for the SE Zagros (McQuarrie, 2004; Motamedi et al., 2012; Pirouz et al., 2017; Najafi et al., 2021)."*

*Line#285: This is exactly what we proposed in Mouthereau et al.(2006, 2007).*

*We mention the reference in the model setup where we introduce the inherited faults.*

*Line#295: Perhaps compare with the newtonian salt viscosity we modelled in our 2006 paper.*

*Thanks for the suggestion. We have added the comparison of studies in discussion section:*

*"Furthermore, Mouthereau et al. (2006) determined that the viscosity of Hormuz salt decreases significantly with increasing temperature, exhibiting Newtonian fluid behaviour under low stress and strain-rate conditions. In our results, the average viscosity of non-Newtonian salt is affected by both temperature and strain rate, where weak zones with the lowest strain rates and temperatures exhibit the largest viscosity*

*values. The resulting values are slightly larger than what was proposed for the Farc arc resulting from Critical Wedge Modelling (Mouthereau et al., 2006)."*

Line#517-520: *This echoes to my previous comments that previous studies of the Fars arc should be better introduced … in the introduction.*

*We added the study of Mouthereau et al (2006) in introduction section:*

*"Mouthereau et al. (2006) reported that basement deformation and thickening is crucial in the Zagros Folded Belt to reproduce the topographic growth observed. They furthermore highlight that the early stage of compression in the Zagros foredeep was controlled by the reactivation of pre-existing faults, suggesting that inherited structural features significantly affect current deformation patterns."*

Line#568: *The analysis of topographic wavelengths is developed in the 2006 paper*

*The reference has been corrected.*

Line#592: *Yes but this is observed east of the Fars arc along the Bandar-abbas segment and strictly speaking this is not the centre of the Fars arc where I think this study best applies (or as justified below this applies to the Inner Fars). Precisions are needed here.*

*Thanks for comments. In the central part of the Zagros fold-and-thrust belt in Iran, several diapirs of the Latest Precambrian–Early Cambrian Hormuz Salt are prominently exposed along a series of N–S to NNW–SSE trending right-lateral tear fault systems. The Karebas Fault System (KBFS) is one of these significant fault systems and is associated with five notable salt diapirs. Additionally, there are several NW–SE trending anticlines in the region, which are either intersected by the fault system or terminate against it and the adjacent salt diapirs (Hassanpour et al., 2018). While it is true that notable interactions between faulting and salt diapirs are observed east of the Fars arc, such relationships are also present and significant within the central Zagros, where the Karebas Fault System is situated. The reference has been added.*

Line#636: *add Mouthereau et al, 2007*

*The references have been updated.*

Lines#643-644: *In our analytical (less quantitative definitively) work (Mouthereau et al., 2006) we concluded that diabase might be too weak to reproduce the topography. Do you have an explanation why we have different conclusions ? Did you have a look to the topographic slope of your models ? The base of your box is horizontal but if you consider flexure the deep decoupling level should be inclined towards the load and your topography might be too low, perhaps below sea level, inconsistent with a mountain range.*

*Thanks for the comment. Reported surface taper angles in the Zagros are generally around 0.5°. Our models result in surface angles slightly above 0.4°, which is very comparable. Obviously, the low taper is determined by the occurrence of a salt décollement. Even for intense basement deformation, the overall taper adjusts according to the resistance of the salt horizon, hence to a certain degree swallow the effect of elastic bending. Regarding the conclusion of whether diabase might be too weak or not, in our opinion this refers to the expected depth of the basement detachment zone. In our case, the crust has an initial thickness of 27 km when convergence starts, and the detachment horizon may freely develop depending on temperature, etc. Looking at Figure 15 in Mouthereau et al. (2006) it can be observed that the expected strength (viscosity) at a depth of 30 km is much larger than at 45 km depth, leading to a lower expected taper.*

*"Mouthereau et al. (2006) on the other hand concluded that diabase might be too weak to reproduce the observed topography in the Fars arc. However, they apply a crustal thickness of 45 km, which implies increased temperature conditions and thus a weaker diabase detachment."*

**##*General comments:**

*I am not an expert in the regional geology of the Zagros fold-and-thrust belt, which is why I can only comment on the modeling aspects of the study.*

*We would like to thank Reviewer #2 (Lorenzo Giuseppe Candioti) for careful reading and constructive review of the manuscript.*

*The authors present 2D thermo-mechanical (one-way coupled) numerical models of tectonic extension and subsequent compression applied to the Zagros fold-and-thrust belt. The model features pre-existing basement faults and a salt layer that acts as décollement horizon. Varied model parameters include (1) the thickness and rheology of the salt layer, (2) the geometry of pre-existing basement faults, and (3) the horizontal velocity of the basement during convergence. Generally, the study is well written and logically organized. However, I think the current version of the manuscript needs further editing. In particular, (1) some aspects could be discussed in more detail and (2) the research question and main contribution of the study could be presented in a clearer way.*

*Introduction: The main insights from previous studies on thin- and thick-skinned tectonics are summarized and introduced well. In addition, the work of Kiss et al. 2020, Spitz et al. 2020, and Humair et al. 2020 could be introduced here as well. They also presented 2D and 3D geodynamic models that highlight the importance of tectonic inheritance for the evolution of fold-and-thrust belts. I would like to read about the observations that all these previous models did not capture. This would help putting this study into perspective.*

*Thanks for comment. We added a new paragraph to the introduction:*

*"Various 2D numerical modeling studies have investigated the evolution of fold-and-thrust belts and salt-bearing basins. Nilforoushan et al. (2013) demonstrated the influence of geothermal gradients and basement mineralogy on fault geometry and basement reactivation in the Fars arc, emphasizing the role of weak salt horizons in mechanical decoupling. Heydarzadeh et al. (2020) analyzed factors such as sedimentation rates, erosion, and salt layer properties in Dehdasht basin, highlighting the importance of balanced surface processes and deformation rates. Humair et al. (2020) conducted simulations to study the interaction of folding and thrusting during Swiss Jura and Canadian Foothills fold-and-thrust belt evolution, focusing on the effects of layer-parallel shortening and initial geometrical perturbations. Their work showed that the magnitude of these perturbations influences whether folding or thrusting predominates, affecting the structural evolution and asymmetry of anticlines. Spitz et al. (2020) conducted 3D thermo-mechanical numerical simulations to investigate the influence of laterally variable inherited structures on fold-and-thrust belt evolution and nappe formation on Helvetic nappe system. The study demonstrated the fundamental importance of tectonic inheritance on fold-and-thrust belt evolution, with strain localization, folding, and nappe transport controlled by initial geometrical and mechanical heterogeneities. Almost all studies have focused on examining the collisional phase and deformation resulting from compression in the fold belts and the Fars Arc, while the earlier extensional history and its effect on later deformation have received less attention (e.g., Granado and Ruh, 2019)."*

*Geological setting: Not being an expert in the regional geology of the Zagros, it seems to me that a lot of research has been conducted on the study area. What could be more focussed on is what exactly remains poorly understood about the geological evolution of the Fars Arc. Also, the particular geologic problem this study is addressing should be clearly outlined in an*

*additional paragraph, and how the presented models will help gaining new insights.*

*Thanks for the comment. In this part of the paper, we introduce the geological setting of the Fars arc and mention the tectonic history of the Zagros fold-thrust belt and the stratigraphic column of the study area based on previous studies. As you suggested, we added an additional paragraph at the end of the geological setting section concerning*

*your comment:*

*"Many studies have been conducted in the Zagros fold-thrust belt and the Fars arc. However, several critical aspects of the geological evolution of the Fars Arc remain poorly understood. Specifically, the precise mechanisms and timing of basement involvement, the interaction between basement faults and salt décollements during*

*tectonic inversion, and the relative contributions of thin-skinned versus thick-skinned tectonics to the overall structural evolution are not fully resolved (see Mouthereau et al., 2006; 2012). We employ a numerical model that simulates the complete tectonic history of the Fars Arc, including an initial extensional phase followed by a compressional phase."*

*Results: I think it would help the reader to interpret the results, if the figures were larger. In the current state, I find it challenging to identify all the important details the models seem to predict. It might also be interesting to see stress, temperature, or viscosity fields for at least the reference model. This would make it a lot easier to identify rheological boundaries and*

*structures.*

*Thanks for pointing this out. We request the journal to input the figures in landscape, to make them better visible. Furthermore, we added a new figure 5, where viscosity and stress of the reference model are discussed individually:*

*"Patterns of the second invariant of the stress tensor after the extensional phase and*

*after full convergence indicate an increase in stress with depth down to ~30 km (y = 50 km), where the brittle-to-ductile transition begins (Fig. 5a,d). The lower part of the basement displays low stresses given its lower viscosities and ductile nature (Fig. 5b,e). Viscosity plots furthermore indicate the position of the low-viscous décollement and basement thrusts."*

*Discussion: Agreements between the presented results and previous studies are discussed well. This suggests that the models are capable of making some realistic predictions for the evolution of the Fars Arc. I think the discussion would benefit from highlighting the advantages of the presented models compared to previous models and how they help gaining new insights into*

*the evolution of fold-and-thrust belts in general. For example, one novel aspect of the models presented here seems to be that rifting is modeled prior to collision. I would like to read some*

*discussion on that particular model feature. Why is it important and what advantages does it bring compared to models that only focus on collision?*

*Thanks for the comment, we have added a paragraph for clarification of the impact of rifting phase in fold thrust belt numerical modeling:*

*"The consideration of rifting prior to collision represents a new aspect that is commonly ignored in previous studies related to the Zagros fold-and-thrust belt. This approach provides a more comprehensive understanding of tectonic evolution by allowing the investigation of pre-collisional structural configurations and their influence on subsequent deformation patterns. Including rifting before convergence is crucial as it sets the initial conditions that significantly impact structural evolution during collision (Buiter and Pfiffner 2003; Ruh et al., 2018; Granado and Ruh, 2019). This approach offers several advantages: It provides insight into structural inheritance, as rifting creates pre-existing weaknesses and fault systems that play a crucial role during subsequent compressional phases. Our model demonstrates how these inherited structures influence strain localization and deformation styles, providing insights into the evolution of complex geological features such as the Fars Arc."*

**##Specific comments:**

*Lines 71-76: Seems like more recent studies favor the latter hypothesis. Are both hypotheses still equally supported by all the data collected so far?*

*We modified this sentence with respect to a comment by reviewer #1:*

*"In the Fars Arc, the activity of inherited faults has affected the progression of deformation towards the foreland, with the Mountain Front Fault being related to basement thrusting (Bahroudi and Koyi, 2003; Mouthereau et al., 2006; Mouthereau et al., 2007a; Yamato et al., 2011; Ruh et al., 2014; Najafi et al., 2021)."*

*Lines 80-82: It would help the reader to better understand the general relevance and importance of this study, if it was clearly stated what is poorly understood and why it is crucial to close this knowledge gap, here. Especially, since the previous paragraph outlined a certain degree of agreement in the community on the role of the décollement layer and the basement faults in that region.*

*We have added a paragraph for clarification of the impact of rifting phase in fold thrust belt numerical modeling:*

*"Understanding these processes is crucial for improving our geological models of the region, which has significant implications for hydrocarbon exploration. Although there is a general consensus on the importance of the décollement layer and basement faults, the detailed dynamics and their broader impact on regional tectonics require further investigation in during inversion tectonics."*

*Line 181: Is the material density assumed to be constant?*

*Yes, we ignored density changes due to thermal expansion or compression because their impact in the crust is minimal in the absence of phase transformations. We mentioned it at the corresponding place in the text.*

*Line 185: Why are contributions from viscous dissipation and radiogenic heat excluded? Are material conductivity and heat capacity constants? Their values do not seem to be provided.*

*These additional heat production mechanisms in the crust are indeed not included in our current model. The relatively short numerical runtime of our modeling means the cumulative effect of these heat sources would be minimal. These parameters are set as constants for each rock type in our model. Our primary interest is in the mechanical deformation, where small variations in thermal properties would have minimal impact on the overall results. For most rock types, we use a thermal conductivity of 2.5 W/(m·K) and a heat capacity of 1000 J/(kg·K), which are typical values for crustal rocks. We added*

*the values in Table 1 for clarity.*

*Line 194: typo -> visco-elasto-plastic/brittle*

*Thanks, has been corrected.*

*Line 196-199: The rotational and advection terms are missing in the objective derivative. How is rigid body rotation included in the model? This is quite crucial for folding simulations (see Schmalholz et al. 2001).*

*The numerical model in this study employs a simplified form of the objective co-rotational time derivative for visco-elastic stresses, as shown in equation (5):*

$$\frac{D\tau_{ij}}{Dt} = \frac{\tau_{ij} - \tau_{ij}^{old}}{\Delta t}$$

*This simplification is a common practice in geodynamic modeling, as seen in works by Gerya and Yuen (2007) and Moresi et al. (2007). While this form does not explicitly show the rotational and advection terms, it is still capable of capturing the essential physics of visco-elastic deformation in many geological settings.*

*We added the references in the text.*

*Line 200: Where does the prefactor 0.5 come from? Is this an additional weakening factor? I suggest calling this viscosity eta_dis or eta_visc, for clarity. Also, shouldn't there be a factor that accounts for the conversion of the experimental 1D flow law to a flow law for stress tensor*

*components?*

*The 0.5 factor is applied as a simplification, independent of the specific deformation type (e.g., shear, plane strain). The equation for effective viscosity is derived from a commonly used form in laboratory experiments: $\dot{\varepsilon} = A.\sigma^n.exp\left(\frac{-Q}{RT}\right)$. With the assumption of $\sigma =$*

*$2\eta\dot{\varepsilon}$ (as we use the second invariant,i.e. deviatoric components), this equation converts to*

$$\eta = 0.5 \cdot \frac{1}{A_D} \cdot \sigma_{II}^{(1-n)} \cdot exp\left(\frac{Q}{RT}\right)$$

*We use $\eta_{disl}$ in the equations.*

*Line 206: Is delta_t_e the Maxwell relaxation time? Why is it set to 1000 yrs? Why isn't the physical time step used here?*

*The selection of $\Delta t_e$ at 1000 years is typically a compromise. It is selected based on the need for numerical stability, accuracy, and the ability to capture the stress history effectively over geological timescales.*

*Using the physical time step directly for updating visco-elastic stresses would not be practical. The physical time step in numerical models is often chosen based on the smallest relevant timescale for the problem at hand, which can be much shorter than the timescales over which elastic effects are significant. The use of a separate elastic time step, rather than the physical time step of the simulation, allows for a controlled implementation of elasticity in the model.*

*We added the information "Maxwell timestep" in the text.*

*Lines 207-209: Z should be dimensionless, but it is not. Where does this formulation come from and why is it used?*

*Thanks, It was a typo in the numerator, we mistakenly used a plus sign instead of multiplication. Has been corrected.*

$Z = \frac{\Delta t_e \cdot G}{\eta + \Delta t_e \cdot G}$ .

*Z is derived from rheological studies, particularly those involving Maxwell-type visco-elastic models (e.g., Gerya and Yuen, 2007, Moresi et al., 2007, Moresi et al.,2003). This parameter is crucial for interpolating between the elastic and viscous responses of the material within a given time step, thus ensuring numerical stability and consistency in stress updates. The references have been added.*

*Line 212: There are brackets missing in the equation. Eta_num does not seem to be used in any other equation noted here. Where is it used in the algorithm?*

*Thanks, It was a typo in the numerator, we mistakenly used a plus sign instead of multiplication. Has been corrected.*

$\eta_{num} = \eta_{disl} \cdot Z = \frac{\eta_{disl} \cdot \Delta t_e \cdot G}{\eta_{disl} + \Delta t_e \cdot G}.$

*The term $\eta_{num}$ is introduced as a numerical (visco-elastic) viscosity, derived from the viscous viscosity $\eta_{disl}$, which is important for solving the set of equations governing these processes and is used to stabilize the numerical solution by adjusting the effective viscosity based on the time step and the plastic potential. $\eta_{num}$ would be integrated into this framework to ensure that the visco-elasto-plastic behaviour is accurately represented (Gerya and Yuen, 2007).*

*Lines 218-219: Sigma_xx and Sigma_xy have not been introduced before, are those components of the total stress tensor? In this formulation it seems that stress tensor components are increased if stresses are below the yield criterion. If this formulation is only valid in case of yielding, maybe using the mathematical notation of cases (curly brace) in the equation would make this formulation clearer.*

*"The components of stress ($\sigma_{xx}$is the normal stress component, and $\sigma_{xy}$ is the shear stress component) and the viscosity are then updated as:"*

*Also, we presented the equation using the mathematical notation of cases (curly braces):*

$$\sigma_{xx}^{new} = \begin{cases} \sigma_{xx}\frac{\sigma_y}{\sigma_{II}}, \; if \; \sigma_{II} > \sigma_{yield}, \\ \sigma_{xx}, \; if \; \sigma_{II} \leq \sigma_{yield} \end{cases},$$

$$\sigma_{xy}^{new} = \begin{cases} \sigma_{xy}\frac{\sigma_y}{\sigma_{II}}, \; if \; \sigma_{II} > \sigma_{yield} \\ \sigma_{xy}, if \; \sigma_{II} \leq \sigma_{yield} \end{cases},$$

*Line 223: This seems non-standard, especially for power-law rheology. Why are calculations not performed on the Eulerian grid?*

*This approach is advantageous for complex rheologies and geometries, such as power-law behavior, because it allows for an increased resolution tracking of material properties and interfaces. The flexibility of the Lagrangian markers ensures that the non-linear and history-dependent nature of visco-elasto-plastic materials is accurately captured. To us this has been standard so far.*

*Line 225: Formatting. 10^25 appears as 1025*

*Thanks, has been corrected.*

*Lines 234 f.: How are basement faults parameterized in the models?*

*Thanks for pointing this out. We added another line in Table 1 listing the parameters of the basement faults.*

*Table 1: What is the underlying assumption for choosing a fluid pressure ratio of 0.4? Values for Cohesion seem to be very low even before softening. What was the motivation to choose*

*such low values and to reduce them further as a function of strain? Why are the sediments parameterized by the same flow law but different densities and friction angles?*

*The choice of a fluid pressure ratio (λ) of 0.4 in the model is based on typical values for hydrostatic fluid pressure as a first-order approximation. The rheological properties of the sediment are based on quartzite. According to the stratigraphic column of the Fars Arc, the nature of the sediments layers varies depending on the tectonic history. These sediments include evaporites and shales, which are weaker than clastics. Either way,*
*these rocks mainly deform in a brittle manner.*

*Lines 243-245: Between the extension and the convergence period in the Zagros there seems to be a 180 Myr period of inactivity which can have an impact, especially on the thermal field and the dynamics in the asthenosphere below, which may in turn have control on the lithospheric deformation. We have shown this in a series of publications (starting with Candioti*
*et al. 2020). As far as I understand, the model switches from extension to convergence instantly. Why is the rifting period included but the passive margin period excluded in the models presented here?*

*We ignored the phase of tectonic quiescence to simplify the modelling approach. We added a sentence in the corresponding section:*

*"The phase of tectonic quiescence between rifting and convergence is ignored here for simplification of the model setup."*

*Lines 371-375: A figure showing the stress and temperature field of the described models would be helpful to support the line of argumentation here.*

*The corresponding line were removed during the revision process.*

*Line 461-462: This is likely a result of the one-way thermomechanical coupling and the strain weakening. I suspect that this frictional weakening of already weak lithologies promotes immediate material failure under compression. In that case, visco-elastic stresses cannot be*
*build up to significantly high values and then be released when shear zones form. I would also not expect to see this effect in Fig 10e. Instead, this may explain the signal pattern (if $vx\_b <$ $vx$) in Fig. 11: In absences of stress built-up and release, the only signal recorded is crustal thickening. As the belt grows, more and more force is necessary to drive the convergence at constant speed. I suspect that if the lithologies were stronger and shear heating would be*
*considered, stresses would build up to higher values and then drop once a shear zone forms. This might then also be visible in at least Fig. 11 (compare to Fig. 11b in Candioti et al. 2021). How do the values for forces compare to estimates for collision zones in general?*

*As stated by the reviewer, it is a competition between vertical growth increasing strength and weakening of faults decreasing it, as outlined in the text. As for a comparison, there*
*is not much information on fold-and-thrust belts, and in mantle-scale experiments, the mantle lithosphere adds to boundary force by a large part. However, we added a reference from analogue modelling comparing the geometry of the temporal evolution of total energy consumed:*

*"A similar pattern of boundary force evolution was also observed from analogue models, however, with more acute variations for specific internal deformation of the compressed sand pile (McBeck et al., 2018)."*

*Line 491: Missing word „of"*

*Thanks, has been corrected.*

*Line 497: Missing word „to"*

*Thanks, has been corrected.*

*Lines 595-596: The diapir in the rifting model is hardly visible. An enlargement or generally larger figures would help identifying the diapirs in the models. A brief discussion about earlier work (e.g., Fernandez & Kaus 2014) would be suitable here.*

*Thanks for spotting this. The figures will be in landscape orientation and therefore better visible. Thanks for suggestion, the studies are discussed:*

*"Fernandez and Kaus (2014) showed with numerical modeling that pre-existing salt diapirs can significantly influence the pattern and growth of three-dimensional folds and fold patterns, accelerating fold formation and localizing deformation, highlighting the important role of diapirism in structural evolution during tectonic processes."*

*Figure 12: I have the impression that the models are generally dominated by faulting whereas the reconstruction seems to show more folding dominated deformation. It would be interesting to see a movie that shows the folding and thrusting in one of these models.*

*That is a good point. We added now GIF movies of all experiments in the Supplementary Material which can be accessed from "Data availability" section of the Manuscript.*

*The GIF movies of all experiments that support the findings of this study are available in Figshare with the identifier: https://figshare.com/s/38141397f97519f7dc31.*

*Line 643-644: The depth of the brittle-ductile transition does not only depend on material parameters and the temperature, but also on strain rate (stress) among other variables. Depending on local conditions, this depth can vary for the same material parameters. It should therefore be generally possible to get similar depths of the brittle-ductile transition for different material parameters at different conditions. Hence, it might not be the best justification for the choice of flow law parameters here.*

*Thank you for the comment. We have revised this part of the discussion accordingly:*

*"By examining the deformation within the basement and the earthquake depths, we estimate the transition zone from brittle to ductile behaviour at around 30 km. While this*

*depth is influenced by multiple factors, including material parameters, temperature, and strain rate (stress), our approximation is based on typical conditions relevant to the region. This suggests that diabase rheology, combined with the applied geotherm, is appropriate for modeling the basement under these specific conditions. Mouthereau et al. (2006) on the other hand concluded that diabase might be too weak to reproduce the observed topography in the Fars arc. However, they apply a crustal thickness of 45 km, which implies increased temperature conditions and thus a weaker diabase detachment.”*

*Figure 13: A description of panel d is missing.*

*Thanks, the figure caption has been corrected.*

*“Figure 13: (a) Seismicity of Zagros fold-and-thrust belt reported by ISC during the years 2000 to 2023 with magnitudes ML> 4 and local temporary network data by Tatar et al (2004) with magnitudes ML<4 , superimposed on a shaded relief map derived from Global Topography. (b, c, d) Projection of earthquakes along profile A–B after removing fixed depths on geological cross section and the reference model (Model 1) after 15 Myr of convergence.”*

*References:*

*Candioti, L. G., Schmalholz, S. M., & Duretz, T. (2020). Impact of upper mantle convection on lithosphere hyperextension and subsequent horizontally forced subduction initiation. Solid Earth, 11(6), 2327-2357.*

*Candioti, L. G., Duretz, T., Moulas, E., & Schmalholz, S. M. (2021). Buoyancy versus shear forces in building orogenic wedges. Solid Earth, 12(8), 1749-1775.*

*Fernandez, N., & Kaus, B. J. (2014). Influence of pre-existing salt diapirs on 3D folding patterns. Tectonophysics, 637, 354-369.*

*Kiss, D., Duretz, T., & Schmalholz, S. M. (2020). Tectonic inheritance controls nappe detachment, transport and stacking in the Helvetic nappe system, Switzerland: insights from thermomechanical simulations. Solid Earth, 11(2), 287-305.*

*Humair, F., Bauville, A., Epard, J. L., & Schmalholz, S. M. (2020). Interaction of folding and thrusting during fold-and-thrust-belt evolution: Insights from numerical simulations and application to the Swiss Jura and the Canadian Foothills. Tectonophysics, 789, 228474.*

*Schmalholz, S. M., Podladchikov, Y. Y., & Schmid, D. W. (2001). A spectral/finite difference method for simulating large deformations of heterogeneous, viscoelastic materials. Geophysical Journal International, 145(1), 199-208.*

*Spitz, R., Bauville, A., Epard, J. L., Kaus, B. J., Popov, A. A., & Schmalholz, S. M. (2020). Control of 3-D tectonic inheritance on fold-and-thrust belts: insights from 3-D numerical models and application to the Helvetic nappe system. Solid Earth, 11(3), 999-1026*

*We integrated the listed references into the manuscript.*

---

## Referee Report (RR1)

I thank the authors for considering my comments and implementing them in the second version of the manuscript. Compared to the previous version, the current manuscript is much clearer. Also the GIFs really help interpreting the evolution of the models. Except for some minor aspects, I have no further suggestions.

**General comments**

The newly added paragraphs seem to be stitched into the manuscript and sometimes interrupt the reading flow. I think it would be worth refining the transitions and maybe reordering the paragraphs for a more seamless integration into the manuscript. An example is given in my specific comments below.

The difference between models that do not include a rifting phase and those presented here could still be discussed in more detail. If modeling the rifting phase is important for the convergence model, what do the presented models predict that pure convergence models with prescribed weak zones fail to predict?

A brief description of what happens in each GIF in the supplement would help the reader to interpret the evolution of the models.

**Specific comments**

line 14: fold-thrust belt (remove the whitespace)

line 15: fold-and-thrust belts or fold-thrust belts (as in line 10)? Please, keep consistent nomenclature for clarity.

line 17: fold-and-thrust systems maybe?

line 101: I would remove „On the other hand" and combine this paragraph with the previous.

line 105: It would be interesting to read more about the potential of hydrocarbon exploration in the Zagros (also mentioned in the abstract, but I didn't read much about it in the manuscript) here and read a discussion on your findings in that context later in the manuscript.

line 125: After introducing the abbreviation, use ZFTB throughout the rest of the manuscript.

line 268: As far as I understand, the basement faults formed during the Permian-Triassic rifting phase, i.e. they should result from the rifting model and serve as initial condition for the convergence model. Why are they already present in the initial configuration of the rifting model?

lines 281-282: It should at least be acknowledged that the quiescence period has an impact on the thermal field (as mentioned in my comments), because the temperature influences the rheology which is an important aspect of this study. Why is it justified to make this simplification?

line 310: Kappa has been used in the temperature equation. For clarity, it would be better to either use different symbols or subscripts for Kappa in these two equations.

lines 454-456: An indication of coordinate would help to identify the pop-up structure.

line 477: This is confusing: is it 25% or 75% of vx?

Figure 11: I guess the y-axis in panel e should have labels of 0, 5, 10, and 15 TN/m instead of 0, 5, 1, 15 TN/m.

line 525: Use either illustrates or shows.

line 594: How much larger is „slightly larger"?

line 616: Exhibited or demonstrated?

lines 674-677: This paragraph disrupts the reading flow. I would move it to the end of this section.

Table 1: In the column header of thermal conductivity, the unit of power should be a capital letter. I guess the unit of cohesion (C) should be MPa instead of MPs.

GIFs: I could not find references to the GIFs in the manuscript. They should be referenced in the text, otherwise the reader has no context when watching them.

---

## Author Response (AR2)

I thank the authors for considering my comments and implementing them in the second version of the manuscript. Compared to the previous version, the current manuscript is much clearer. Also the GIFs really help interpreting the evolution of the models. Except for some minor aspects, I have no further
5   suggestions.

> *We would like to thank Reviewer #2 (Lorenzo Giuseppe Candioti) for careful reading and a constructive review of the manuscript.*

**General comments**

10  *The newly added paragraphs seem to be stitched into the manuscript and sometimes interrupt the reading flow. I think it would be worth refining the transitions and maybe reordering the paragraphs for a more seamless integration into the manuscript. An example is given in my specific comments below.*

> *Thanks, we tried to rewrite and move the mentioned paragraph connections and flow*
15  > *accordingly.*

[revised manuscript text omitted]

*The difference between models that do not include a rifting phase and those presented here could still be discussed in more detail. If modeling the rifting phase is important for the convergence model, what do the presented models predict that pure convergence models with prescribed weak zones fail to predict?*

*During tectonic inversion, faults invert from a normal to a reverse mechanism. In the model, when the extension phase is considered, the faults that were affected by extension are reactivated in reverse. This highlights the significant influence of the earlier extension phase on the structures and localization of strain.*

*in the manuscript, we have already mentioned to this important. (for example, in the Comparison with numerical modelling studies section).*

85    *A brief description of what happens in each GIF in the supplement would help the reader to interpret the evolution of the models.*

> *Thanks, for suggestion. We extended the supplementary word file with explanations related to each of the presented GIF.*

90    **# Specific comments**

*line 14: fold-thrust belt (remove the whitespace)*

> *Thanks, we have changed all to "fold-and-thrust belt".*

*line 15: fold-and-thrust belts or fold-thrust belts (as in line 10)? Please, keep consistent nomenclature for clarity.*

95      > *Thanks, we have changed all to "fold-and-thrust belt".*

*line 17: fold-and-thrust systems maybe?*

> *Thanks, corrected.*

*line 101: I would remove „On the other hand" and combine this paragraph with the previous.*

> *Thanks, has been removed.*

100   *line 105: It would be interesting to read more about the potential of hydrocarbon exploration in the Zagros (also mentioned in the abstract, but I didn't read much about it in the manuscript) here and read a discussion on your findings in that context later in the manuscript.*

> *Thank you for your comment. However, the aim of our study is not to discover areas with hydrocarbon potential. Our reference to it was just to point out one aspect of our study,*
105    > *which is that by examining structural evolution, it is possible to assist in identifying hydrocarbon potential. We prefer not to add another paragraph.*

*line 125: After introducing the abbreviation, use ZFTB throughout the rest of the manuscript.*

> *Thanks, we have changed all to "ZFTB".*

*line 268: As far as I understand, the basement faults formed during the Permian-Triassic rifting*
110   *phase, i.e. they should result from the rifting model and serve as initial condition for the convergence model. Why are they already present in the initial configuration of the rifting model?*

> *Yes, that is correct. We want to model the rifting phase where inherited faults have a certain the position in the model as an initial condition. Furthermore, no inherited faults*
115    > *lead otherwise to no localized normal faults in the model during the rifting phase (see the Model 7 in our tests).*

*lines 281-282: It should at least be acknowledged that the quiescence period has an impact on the thermal field (as mentioned in my comments), because the temperature influences the rheology which is an important aspect of this study. Why is it justified to make this*
120   *simplification?*

*We simplified the model by assuming that the temperature field reaches a steady state by the beginning of the collision phase. This approach takes the essential thermal conditions resulting from the quiescence period without significantly complicating the model.*

*"Tectonic quiescence affects the thermal state and, consequently, the rheological behaviour of the rocks. However, we assume that the temperature field reached a steady state by the onset of the collision phase to simplify the model setup. This implementation allows us to effectively capture the relevant thermal conditions while maintaining focus on the critical dynamics of the extension and collision phases."*

*line 310: Kappa has been used in the temperature equation. For clarity, it would be better to either use different symbols or subscripts for Kappa in these two equations.*

*Thanks, it has been corrected, we use K now.*

*lines 454-456: An indication of coordinate would help to identify the pop-up structure.*

*Thanks for suggestion, we added the x coordinate:*

*"A large pop-up structure (x ≈ 230 km) without mechanical decoupling between the basement and the sedimentary cover develops in front of the salt pinch-out"*

*line 477: This is confusing: is it 25% or 75% of vx?*

*The sentence has been changed. "With a basement involvement of 25% ($v_{x\_b} = 0.75 \cdot v_x$), the fault on the right side almost entirely left the model domain, and the other two faults experience lesser degrees of involvement, roughly displaying 100% of tectonic inversion (Model 10; Fig. 10c)."*

*Figure 11: I guess the y-axis in panel e should have labels of 0, 5, 10, and 15 TN/m instead of 0, 5, 1, 15 TN/m.*

*Thanks, it has been corrected.*

*line 525: Use either illustrates or shows.*

*Thanks, it has been corrected.*

*line 594: How much larger is „slightly larger"?*

*We changed it to "larger".*

*line 616: Exhibited or demonstrated?*

*Thanks, it has been corrected.*

*lines 674-677: This paragraph disrupts the reading flow. I would move it to the end of this section.*

*Thanks for the suggestion, we moved the paragraph.*

*Table 1: In the column header of thermal conductivity, the unit of power should be a capital letter. I guess the unit of cohesion (C) should be MPa instead of MPs.*

*Thanks, we corrected the mentioned errors.*

*GIFs: I could not find references to the GIFs in the manuscript. They should be referenced in the text, otherwise the reader has no context when watching them.*

*Thanks for the comment. We now mention all GIF's presented in the Supplementary Material in the main text when applicable as Fig. S1-11.*

160

---

## Author Response (AR3)

Dear authors,

Before i can accept the manuscript for publication, i need to ask you for another careful check
regarding the description of the numerical model and the modelling approach. Please see my
comments below, but i may well still have missed some points and would thus ask you to read
especially section 3 critically.

*We would like to thank the editor (Susanne Buiter) for careful reading and a constructive
review of the manuscript.*

The question to keep in mind is if enough information is provided for someone to rerun the models.

*To the best of our knowledge, we provide all necessary information on geometry,
boundary conditions, and mechanical and rheological constraints to reproduce the
numerical experiments presented here. In either case, we carefully went through the
manuscript again, with focus on section 3, and answered the points raised below to the
best of our knowledge.*

I also support the point made by reviewer #2 as this is an opportunity to clarify an essential
characteristic of your models. I therefore would like you to explicitly address the question raised by
reviewer #2 in your manuscript:
"The difference between models that do not include a rifting phase and those presented here could still
be discussed in more detail. If modeling the rifting phase is important for the convergence model,
what do the presented models predict that pure convergence models with prescribed weak zones fail
to predict?"

*Thanks for mentioning the lack of emphasis on this key feature of the presented models.
Summarized in a simplified manner, including a previous extension phase allows to
investigate the effects of structural and geometric features related to this extensional
phase on the structural and mechanical evolution during compression. Obviously, any
feature may also be implemented in an initial model before shortening without extension,
but the strain-related and structural constraints would be less dynamic. The
implementation of predefined faults helps reconstructing the specific location of inherited
faults in the Zagros. We added a sentence to the introduction and added a paragraph in
the discussion, describing in more detail the importance of a rifting phase for the
obtained results:*

*"Incorporating a rifting phase into the model setup result in more realistic initial
conditions for the convergence stage, featuring variations in crustal thickness, rift-
related sedimentary basins and the presence of weak zones."*

*"An essential characteristic of the numerical experiments presented here is the
implementation of a rifting phase prevailing crustal shortening. Earlier numerical
studies on tectonic inversion demonstrated that strain-related weakening of the crustal
basement during an extensional tectonic phase has an important effect on the localization
of deformation during consequent convergence (Ruh and Vergés, 2018). Furthermore,*

*the mechanical strength of sedimentary deposits filling rift-related basins has been shown to influence the structural evolution during tectonic inversion, where weak syn-rift deposits favour the development of hanging wall by-pass structures (Granado and Ruh, 2019). In contrast, various studies mimic the extensional phase by prescribing inherited structures and geometric configurations comparable to rifted margins instead of conducting an extensional phase when investigating the tectonic inversion of fold-and-thrust belts (e.g. Buiter and Pfiffner, 2003; Nilforoushan et al., 2013; Bauville and Schmalholz, 2015; Kiss et al. 2020). However, the implementation of an extension phase allows for a dynamic formation of basement steps and graben geometry based on the thermo-mechanical model characteristics such as the visco-plastic/brittle transition and the evolution of the salt decoupling along the basement top (Fig. 4a-c and 5a,b). Model 7 demonstrates that a certain way of strain localization is needed to form narrow shear bands similar to basement faults during extension (Fig. 9c). While other studies impose specific boundary conditions (Ruh and Vergés, 2018) or thermal variations (Ruh and Vergés, 2018), we introduced predefined geometries to localize extensional normal faults, controlling the position of basement deformation (Fig. 3b)."*

In addition, could you contrast the following observations (here in the summary from the abstract) for the models with prescribed weak zones with the model in which inherited faults were not present (Model 7)?

"The inverted basement faults form large foreland-verging fault-propagation anticlines in the sedimentary cover, while the thick salt layer promotes the growth of second-order detachment anticlines accompanied by both fore- and back-limb thrust faults."

*Thank you. According to your suggestions, the text has been revised:*

*"Experiments without prescribed basement faults result in dispersed brittle/plastic deformation during rifting and convergence and an effective mechanical decoupling along the salt horizon."*

Model 7 without prescribed weak zones shows a substantial difference to models that have prescribed weak zones. Could you add an explanation for the structural style exhibited in Model 7? My guess is that the salt layer is so weak that it decouples basement shortening from shortening above the salt layer, thus effectively leading to a thin-skinned style in the topmost layer. Fault-propagation folds can however still form and the role of salt is still pronounced. The low rheological strength of the salt layer will also explain the tendency to form conjugate thrusts (pop-ups) rather than vergence in one direction. The topography that is obtained at the base of the salt layer is interesting. Does it form because the lower crust behaves viscous, thus leading to a form of isostatic compensation?

*Thanks for your comment, we added a paragraph describing the specific points raised in the discussion, section 5.1:*

*"All experiments with prescribed weak zones within the basement show intense inversion along these inherited structures (e.g., Fig. 4). A particular experiment is Model 7, where no basement faults were prescribed (Fig. 9c,d). The structural style observed in Model 7 is driven by the low rheological strength of the salt layer, which decouples basement*

*shortening from the overlying layers, resulting in thin-skinned deformation of the cover sequence (Fig. 9d). This weak salt layer promotes the formation of fault-propagation folds and conjugate thrusts (pop-ups) rather than a clear structural vergence, as observed by a large pop-up structure at $x \approx 230$ km. The topography of the top of basement is likely caused by a critical wedge geometry defined by the viscous strength of the lowermost modelled crust."*

**Detailed comments:**

Please explain somewhere that the numerical model generates shear zones, that you refer to as faults (which many will view as a more discrete feature).

*Thanks for the comment. We have mentioned to this point in the initial paragraph of the results section:*

*"While the numerical model is based on a viscous formulation and thus develops localized shear zones, we refer to them as faults if they form due to the implemented Drucker-Prager failure criterion (see equation 11)."*

3.1 Governing equations
Please justify why it is appropriate to exclude heat production from the thermal equation for a model that includes the entire continental crust. How representative is a linear geotherm for continental crust?

*Thanks for this comment. There are several reasons for this simplification. One is the geometry and the boundary conditions of the model domain, that impede thermal diffusion and advection. Hence, a simplified temperature description was preferred. We have added more information in the respective section:*

*"Additional heat production, such as radioactive heating and shear heating, is not activated in the presented experiments due to the geometrical constraints of the model setup and related boundary conditions that affect the diffusion of such secondary heat production."*

*The continental crust can exhibit variations due to heat sources like radioactive decay, mantle heat flow, or tectonic activity. Based on our simplification for modeling, a linear initial geotherm is assumed, which is a reasonable approximation for the continental crust:*

*"Applying a linear geotherm is a reasonable simplification for the uppermost ~50 km of the continental lithosphere (Hasterok and Chapman, 2011; Goes et al., 2020)."*

3.2 Rheological model
Line 235, gravitational acceleration g has an index in equation 2, but misses the indes in the symbol explanation. I assume it is the vertical component of gravity. Please correct.

*Thanks, we have changed to "$g_i$" ($g_1 = 0$; $g_2 = 9.81 \frac{m}{s^2}$),*

How important is elasticity for your models? Could you add an estimate of the Maxwell relaxation time and the Deborah number?

*We have added more information about the important of elasticity:*

130 *"Elasticity plays a key role in capturing short-term stress accumulation and release, crucial for fault and fold behavior. The Maxwell model allows the simulation of both immediate elastic response and long-term viscous flow, ensuring that important transient phenomena such as fault reactivation and seismic activity are accurately represented during both extension and convergence."*

135 *Regarding the Maxwell time and the Deborah number, we added the following statements:*

*"Given the numerical time step of $\Delta t_e$ = 1000 years, material undergoing deformation at viscosities below $3.16 \times 10^{21}$ Pa·s can be considered predominantly viscous, while deformation at viscosities above $3.16 \times 10^{21}$ Pa·s is to a significant part elastic and thus*
140 *reversible. Elastic relaxation time varies between ~1 year and 1 million years, depending on the viscosity of the material, which results in Deborah numbers of $10^{-7}$ – 0.1 for a deformation period of 10 million years."*

Table 1: please add the value for shear modulus G

145 *Thanks, we have added the value of G in the text because it is constant for all markers type:*

*"G is the elastic shear (100 GPa for all materials here)".*

Please describe how strain weakening of angle of internal friction phi and cohesion c is achieved.

150 *We added a sentence in the corresponding section:*

*"Strain weakening is implemented by a linear decrease of the values for frictional angle ($\varphi$) and cohesion (c) between a lower and upper strain threshold, defined by $\varepsilon_w^0$ = 0.1 and $\varepsilon_w^1$ = 1."*

155 3.3 initial geometrical setup
Please give the justification for using a 30 km thick crust.

*The continental crust in the Zagros is 40 – 45 km thick (Jimenez-Munt et al., 2012; Taghizadeh-Farahmand et al., 2015). However, given the model setup with the induced shear zone (velocity boundary condition) at the base, we are interested in the thickness*
160 *of the deformed basement. We added a sentence in the corresponding section:*

*"The initial marker distribution defines, from bottom up, 1) a 30-km-thick crustal basement layer, given the depth of basement crustal detachment (Vergés et al., 2011; Kendall et al., 2019)"*

165 "Eulerian cell initially contains 16 randomly distributed Lagrangian markers carrying rock information and properties". The number of markers in a cell will change because of their movement through the grid. Please describe the policy for empty cells and whether a population reduction is applied to cells with many markers.

*We apply a common way to deal with this problem. If no Lagrangian markers are*
170 *available to be interpolated onto a node, the old values of these nodes are used, until a marker moves into their area of influence again. This concerns all parameters that are important to solve the system of equations such as density, viscosity, elasticity, thermal parameters, etc. However, given the large number of Lagrangian markers and the incompressible manner of the resulting velocity field (what goes out goes in), it is highly*
175 *unlikely that a cell remains empty. We added a sentence in the Initial geometrical setup section:*

*"If the finite spatial domains of a specific node become empty of any Lagrangian marker, the previous interpolated parameters are applied for solving the system of equation of this particular node."*

180 And in closing, if the reviews by the two reviewers were useful for your manuscript, and i hope they were, i would appreciate an acknowledgement of the reviewers.

*"We would like to express our sincere gratitude to the reviewers, Dr. Frédéric Mouthereau and Dr. Lorenzo Giuseppe Candioti, for their time and effort in reviewing our paper. Their insightful comments and suggestions have significantly improved the*
185 *quality of the manuscript. Additionally, we extend our appreciation to Dr. Susanne Buiter for her comments and editorial guidance throughout the review process."*

With best wishes,
190 Susanne Buiter